# Spatiotemporal proteomic atlas of multiple brain regions across early fetal to neonatal stages in cynomolgus monkey

Jingkuan Wei[1,2,4], Shaoxing Dai[1,2,4], Yaping Yan[1,2,4], Shulin Li[1], Pengpeng Yang[1], Ran Zhu [1], Tianzhuang Huang[1,2], Xi Li[1,2], Yanchao Duan[1,2], Zhengbo Wang [1,2] ✉, Weizhi Ji [1,2,3] ✉ & Wei Si[1,2,3] ✉

Fetal stages are critical periods for brain development. However, the protein molecular signature and dynamics of the human brain remain unclear due to sampling difficulty and ethical limitations. Non-human primates present similar developmental and neuropathological features to humans. This study constructed a spatiotemporal proteomic atlas of cynomolgus macaque brain development from early fetal to neonatal stages. Here we showed that (1) the variability across stages was greater than that among brain regions, and comparisons of cerebellum vs. cerebrum and cortical vs. subcortical regions revealed region-specific dynamics across early fetal to neonatal stages; (2) fluctuations in abundance of proteins associated with neural disease suggest the risk of nervous disorder at early fetal stages; (3) cross-species analysis (human, monkey, and mouse) and comparison between proteomic and transcriptomic data reveal the proteomic specificity and genes with mRNA/protein discrepancy. This study provides insight into fetal brain development in primates.

The central nervous system undergoes intricate changes that include cell proliferation, differentiation, and assembly into functional circuits and regions during post-implantation stages[1,2]. Neural cells undergo a variety of molecular and morphological alterations during these fetal stages[3,4], including changes related to cerebral and cerebellar differentiation, cortical and subcortical differentiation, and the emergence of specific cortical regions. During this period, any abnormalities may lead to neurological and neuropsychiatric disorders[5-8].

Large-scale omics technology can disentangle the immense complexity of the brain by mapping the connectome and providing a deep molecular characterization of specific brain regions. The resulting omics maps will advance our understanding of brain function and facilitate the study of neurological diseases. Maps of RNA transcriptional dynamics based on RNA abundance, epigenetic regulators, and chromatin accessibility across primate brains have been available since 2011[9-13]. However, accumulating data reveal a poor correlation between mRNA expression and protein abundance[14-17]. Current proteomics analyses of human brain development are only available during postnatal stages[18-21]. There is, therefore, a lack of well-defined stage and region-specific proteomic data covering fetal and neonatal primate brain development. This is due to ethical and sampling limits in humans. However, this gap currently limits our understanding of primate brain development in terms of changes in functional gene products.

Non-human primates share many genetic and physiological similarities with humans, particularly the central nervous system. Due to limited access to healthy human brain tissue, the macaque (an Old-World monkey) brain is an ideal model for the study of brain

[1]State Key Laboratory of Primate Biomedical Research; Institute of Primate Translational Medicine, Kunming University of Science and Technology, 650500 Kunming, Yunnan, China. [2]Yunnan Key Laboratory of Primate Biomedical Research, 650500 Kunming, Yunnan, China. [3]Chinese Primate Biomedical Research Alliance (CPBRA), 650500 Kunming, Yunnan, China. [4]These authors contributed equally: Jingkuan Wei, Shaoxing Dai, Yaping Yan. ✉e-mail: wangzb@lpbr.cn; wji@lpbr.cn; siw@lpbr.cn

development. The neocortical regions of the brain are involved in motor, perceptual, and higher cognitive functions[22], which are highly developed in primates. In macaques, the neocortex makes up 72% of the brain[23], which is close to 76% in humans. Abnormalities in the neocortex are highly correlated with psychiatric and neurodegenerative diseases[24]. Therefore, proteomic studies on macaques can provide a valuable reference for understanding the biological networks, pathways, and dynamic changes in the human brain and shed light on risk genes driving brain disease pathogenesis.

Here, we use a state-of-the-art 4D-Proteomics technology[25] to quantify the protein abundance of 156 samples including 18 brain regions that are anatomically and functionally related to the human brain. Samples were collected across fetal and neonatal stages (nine fetuses and three neonatal cynomolgus macaques). Combining fine anatomical division, dense temporal coverage, and high-resolution proteomics, a dynamic proteomic map of the developing brain with high spatial resolution was obtained. Our data reveal spatiotemporal dynamics in protein abundance during fetal neurodevelopment, the unique pattern of CB, the differential development of cortical and subcortical regions, and changing trends in the protein of risk genes related to neuropsychiatry and degenerative diseases. Our systematic analysis, in a spatially resolved manner, revealed molecular signatures of fetal brain development in nonhuman primates. By comparing macaque proteomics data to previously reported human and mouse brain proteomes[18,26], we identify similarities and differences in these model systems. We also found that there was a certain proportion (>15%) of genes with mRNA/protein discrepancies, which emphasizes the importance of proteomic studies of brain development and is an important complement to previous transcriptomic studies. Our study provides an important resource that provides insight into fetal brain development in primates which may have profound implications for the neuroscience community.

## Results

### Study design and overview of protein profiles across fetal brain development in cynomolgus monkey

The natural gestation of the cynomolgus monkey is about 150 days[27]. We chose four critical developmental time points (four stages) in fetal development: 50, 90, and 120 days post-fertilization (F50, F90, and F120), and postnatal 3 days (P3). These time points encompass peak periods of neurogenesis and cortical expansion during brain development[2]. Estradiol (E2) and progesterone (P4) levels of the pregnant female cynomolgus monkey were continuously measured to accurately determine the developmental stage post-fertilization (Supplementary Fig. 1). High-quality forebrain samples from 12 brains (three biological repeats for each stage) were systematically dissected to sample 18 anatomical brain regions linked to higher-order cognition and behavior, including neocortex (frontal lobe, FL; temporal lobe, TL; Parietal lobe, PL; Primary visual cortex, V1), cerebellum (CB), striatum (STr), hippocampus (Hipp), medial dorsal nucleus of thalamus (MD), and amygdala (Amy) (Fig. 1A). The brain dramatically expands between F50 and birth. There were five regions (PFC, TL, PL, V1, and CB) in F50 brains, 11 regions (aPFC, pPFC, IC, M1, TL, PL, V1, CB, STr, Hipp, and Amy) at F90, and 18 regions (sPFG, mPFG, iPFG, OFC, IC, M1, aCG, sTG, mTG, iTG, S1, sPL, V1, CB, STr, Hipp, MD, and Amy) at the F120 and P3 stages (Fig. 1A and Supplementary Table 1). The protein profiles of the resulting 156 samples were quantified using high-resolution 4D-Proteomics. The protein abundances were expressed as label-free quantification (LFQ) intensity and log2 transformed before differential expression analysis. (Fig. 1B and Supplementary Data 1). In the 156 samples, an average of 5967 proteins were identified per sample (5423–6600 proteins). The union set of identified proteins from the 156 samples was 9286, of which 7618 proteins were annotated with gene information. There was an average of 6002 proteins per stage (5885–6191 proteins) (Fig. 1C and Supplementary Data 2). The

largest and least average number of proteins were identified at F50 and P3, respectively (Fig. 1C). The differentially expressed proteins (DEPs) were calculated by the Student's $t$-test with Bonferroni correction. The number of DEPs in different regions at the four stages are shown in Fig. 1D.

The functions of the cortex, subcortical region, and CB are executed by different proteins distributed at various subcellular localizations. Therefore, we analyzed the dynamic changes of proteins with various subcellular localizations from F50 to P3 among brain regions. We found a high degree of similarity in the proportion changes of protein abundance within subcellular localizations among different regions of the cortex (Fig. 1E and Supplementary Data 3). Nucleoplasm and plasma membrane proteins showed a similar pattern with a general ascending trend in all regions except for the CB. The most abundant cytosol proteins were increased and then decreased in the cortex following development, but showed a continuous decreasing trend in other brain regions.

In addition, we analyzed the patterns of spatiotemporal change of the protein families. There are three major patterns in cortical areas including continuous ascending (such as Protein kinase superfamily and Spectrin family), continuous descending (such as ALB/AFP/VDB family and Intermediate filament family), and descending followed by ascending (such as Tubulin family and Metallo-dependent hydrolases superfamily) (Fig. 1F and Supplementary Data 4). These results elucidate the dynamic changes of several protein families during brain development, which may help to identify key factors in early brain development at the protein level.

### Dynamic proteomic changes across developmental stages and brain regions

PCA analysis of samples showed a clear inverted 'V' trend in the distribution of brain samples across stages for cerebrum samples (Fig. 2A). Samples at the F50 stage were separated by PCA from samples at later stages, and the late fetal stage (F120) was highly similar to the neonatal stage (P3). The correlation analysis revealed that all correlations between any stages were significant ($p < 0.001$), and the highest correlation (0.89) of adjacent stages was observed between F90 and F120 (Fig. 2B). Previous study reported that the variability among developmental stages was greater than those among brain regions during postnatal brain development in human based on proteomic data[18]. In contrast, we found an opposite situation during fetal to neonatal stages in cynomolgus monkey (Fig. 2C). A total of 6177 DEPs were identified across developmental stages, which was higher than that among all brain regions (4856 DEPs). A similar trend was also observed both in cortical and subcortical regions, which indicates more dynamic changes across fetal stages.

Interestingly, further trend analysis revealed that the inter-regional changes of protein abundance exhibited an overall "V" shaped pattern which bottomed at the F90 in CB or at the F120 stage in CTX and sCTX (Fig. 2D). These data suggest that the F90 or F120 stages may be the inflection point for cellular function during brain development. Subsequently, we identified marker proteins (upregulated proteins) for each stage (Fig. 2E and Supplementary Data 5). Proteins encoded by the *MCM2-7* genes were upregulated at the F50 stage. These genes are required for both DNA replication initiation and elongation. APP and MACF1, which play roles in neurite outgrowth and neuronal migration, were upregulated at the F90 stage. In F120 tissue, we saw a higher abundance of CPNE2 and PLPPR3. These proteins function in the regulation of neural stem cells, and in nervous system development and maintenance, respectively. Synaptic proteins STXBP1 and LRRC7, associated with intellectual disability and childhood emotional dysregulation, were upregulated at the P3 stage. Enrichment analysis of these stage marker proteins revealed different biological processes for each stage (Supplementary Fig. 2).

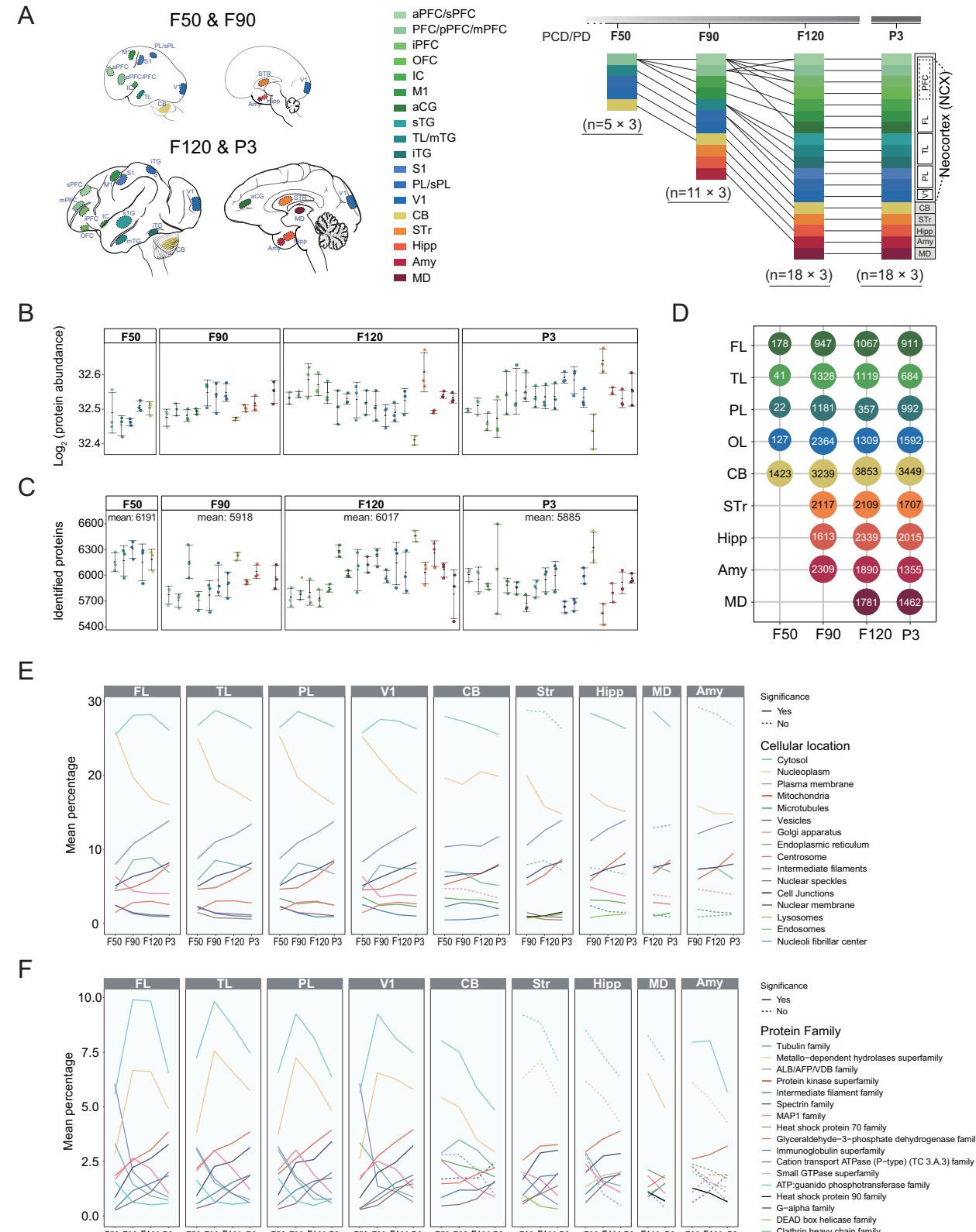

## Proteomic changes in the cerebellum are distinct from the cerebrum

Since the protein profile of CB was distinguished from the cerebrum (Fig. 2A), we further explored proteomic profiles in this region across the four developmental stages. The results showed that protein profiling in the CB was distinct from the other brain regions at each stage (Fig. 3A). The dissimilarity (Euclidean Distance) between CB and the other brain regions (except V1) increased over development (Fig. 3B). By differential expression analysis, we identified 590, 446, 1090, and 1072 proteins that were significantly upregulated in CB compared with the other brain regions at F50, F90, F120, and P3, respectively (Supplementary Data 6). The top 20 marker proteins (upregulated proteins) sorted by AUC value for each stage were shown as heatmaps (Fig. 3C). The CB marker proteins were annotated as distinct functional

**Fig. 1 | Sampling overview and global proteomic profiles across fetal brain development in cynomolgus monkey. A** Schematic showing brain regions and developmental time of samples collected from the monkey brain. A total of 18 regions (aPFC/sPFC, PFC/pPFC/mPFC, iPFC, OFC, IC, M1, aCG, sTG, TL/mTG, iTG, S1, PL/sPL, V1, CB, STr, Hipp, MD, Amy) from four different stages (F50, F90, F120, P3) were profiled by proteomics (left panel). Detailed sampling regions at the four stages are illustrated with different colors (right panel). Each region has three biological replicates from three different animals at the stages shown. **B, C** The total protein abundance (**B**) and the number of identified proteins (**C**) in the 18 regions at the four stages with three biological repeats, respectively. The average number of the identified proteins per stage is shown. The region color is the same as (**A**). The

dot, horizontal line, and error bar indicate the sample value, mean value, and standard deviation, respectively. **D** The number of differentially expressed proteins (DEPs) in different regions and stages is shown. The DEPs in the indicated region were identified by comparing to other brain regions at the same developmental stage. **E, F** The line chart shows the percentage change of the subcellular localizations (**E**) and protein families (**F**) for the identified proteins along with developmental stages in different brain regions. The dashed line indicates no significant difference between any two adjacent stages. The solid line indicates that there is at least one significant difference between the two adjacent stages. The *p*-values were calculated by two-sided Student's *t*-test (**E, F**). Source data are provided as a Source Data file.

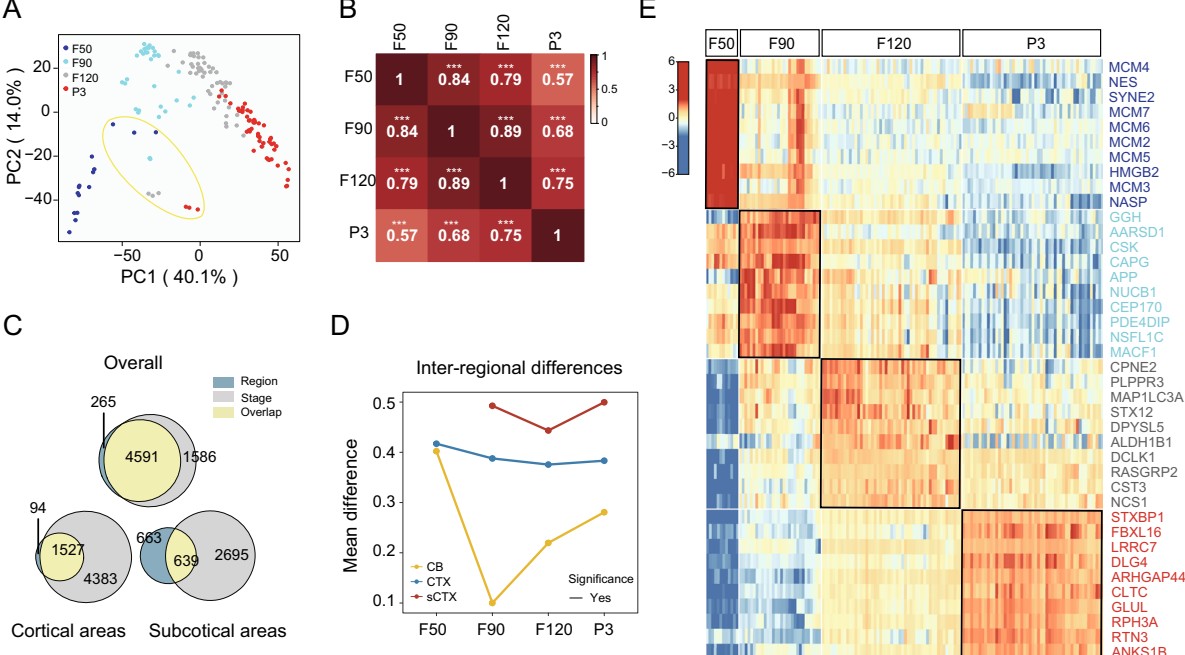

**Fig. 2 | Significant differences between developmental stages are revealed by spatiotemporal proteomics. A** Principal component analysis (PCA) showing the distribution of all samples based on protein abundance. The yellow circle indicates the samples of CB. **B** Correlations between the four stages based on protein abundance. The significance of the correlation is indicated by asterisks (***: *p* < 0.001). *p*-values were calculated by two-sided Student's *t*-test. **C** Venn diagram showing the number of DEPs for stage and region. **D** Line chart showing the inter-regional differences over the stages in regions cerebellum (CB), cortex (CTX), and subcortical tissue (sCTX). The solid line indicates that there is at least one

significant difference between the two adjacent stages. The dashed line indicates no significant difference between any two adjacent stages. The *p*-values were calculated by two-sided Student's *t*-test. Note: three biological repeats of brain regions at each stage were used to calculate the average inter-regional differences between cortical and subcortical areas except that three technical replicates were performed on two CB samples at stage F90. **E** Marker proteins of the indicated stages (F50, F90, F120, and P3). As the CB region is significantly different from the other regions, CB is not included in this panel. Source data are provided as a Source Data file.

pathways during the four developmental stages (Supplementary Fig. 3). At the F50 stage, the CB marker proteins were found to be mainly linked to neurogenesis, and include trans-synaptic signaling, synaptic transmission, and axonogenesis. At the F90 stage, CB marker proteins were enriched in the processes of DNA replication and translation synthesis. The biological processes linked to CB marker proteins at the F120 and P3 stages showed high similarity and were mainly enriched in mRNA processing-related processes. Among these upregulated proteins, GRID2IP and PCP2 are known as the marker proteins of Purkinje cells, and ZIC1 and GABRA6 are markers of granule cells[28,29]. Here, we depict the dynamic changes of the known marker proteins of Purkinje cells and granule cells during fetal development (Fig. 3D, E). The abundance of GRID2IP and PCP2 increased since F50, which indicates the proliferation or maturation of Purkinje cells at this stage. The abundance of ZIC1 increased since F50 and GABRA6 increased lately from F120. In human, granule cells were differentiated from progenitor cells at 10 weeks post-conception, and the GABRA6 as a subunit of GABA-A receptor is highly expressed in the postnatal CB,

which mediates neuronal inhibition[30]. The results suggest that the development of granule cells and the high expression of GABRA6 in the perinatal CB are similar between humans and monkeys. In addition, we analyzed the changes of proteins (APC, CTNNB1, DDX3X, PTEN, BCOR) associated with medulloblastomas in CB at the four developmental stages. We found that the abundances of these proteins were maintained at a constant level with no significant changes between adjacent stages (Supplementary Fig. 4).

## Dynamic proteomic changes and differences between cortical and subcortical regions

As brain development progressed, the difference in protein abundance between cortical and subcortical regions gradually increased (Fig. 4A). At stage F50, the subcortex cannot be anatomically distinguished, so only cortical regions were sampled and measured. From the F90 stage onward, cortical and subcortical regions differentiated along two different paths (Fig. 4A). Therefore, we systematically compared the changes in protein abundance and biological process

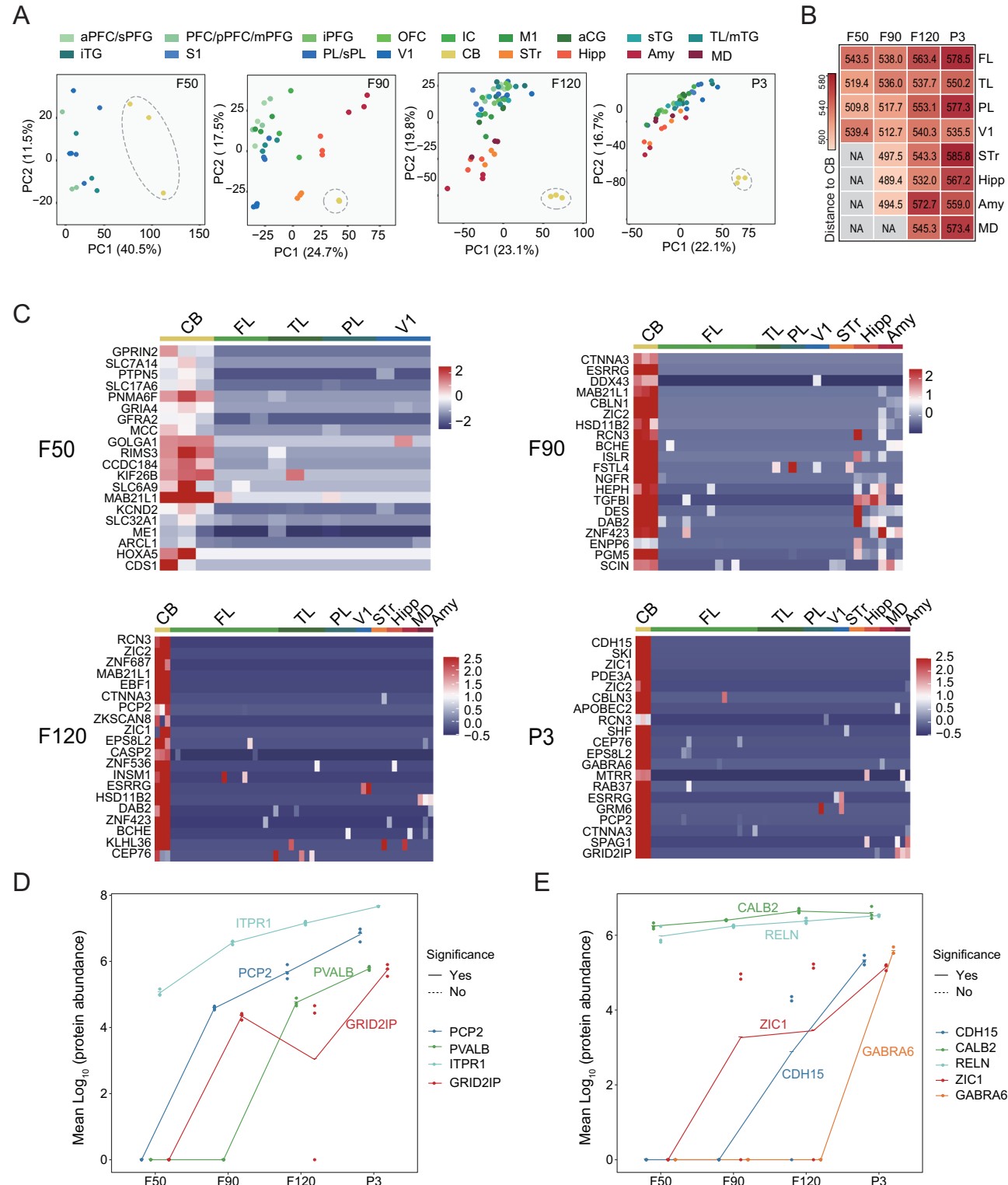

**Fig. 3 | Dynamic proteomic changes in cerebellum. A** Principal component analysis (PCA) showing the CB is distinct from other regions based on protein abundance across the four stages. Note: Three biological repeats were performed on all brain regions of the four stages except CB at the F90 stage. Instead, three technical replicates were performed from two CB samples at the F90 stage. **B** Heatmap showing the Euclidean Distance of CB to the other brain regions at the four stages. **C** Heatmap showing the abundance of the top 20 marker proteins in CB compared to the other brain regions at the four stages (F50, F90, F120, P3).

**D, E** The line chart showing the abundance change of marker proteins for Purkinje cells (**D**) and granule cells (**E**) along with developmental stages in different brain regions. The dot and horizontal line indicate the sample value and mean value, respectively ($n = 3$). The solid line indicates that there is at least one significant difference between the two adjacent stages. The dashed line indicates no significant difference between any two adjacent stages. The $p$-values were calculated by two-sided Student's $t$-test (**D, E**). Source data are provided as a Source Data file.

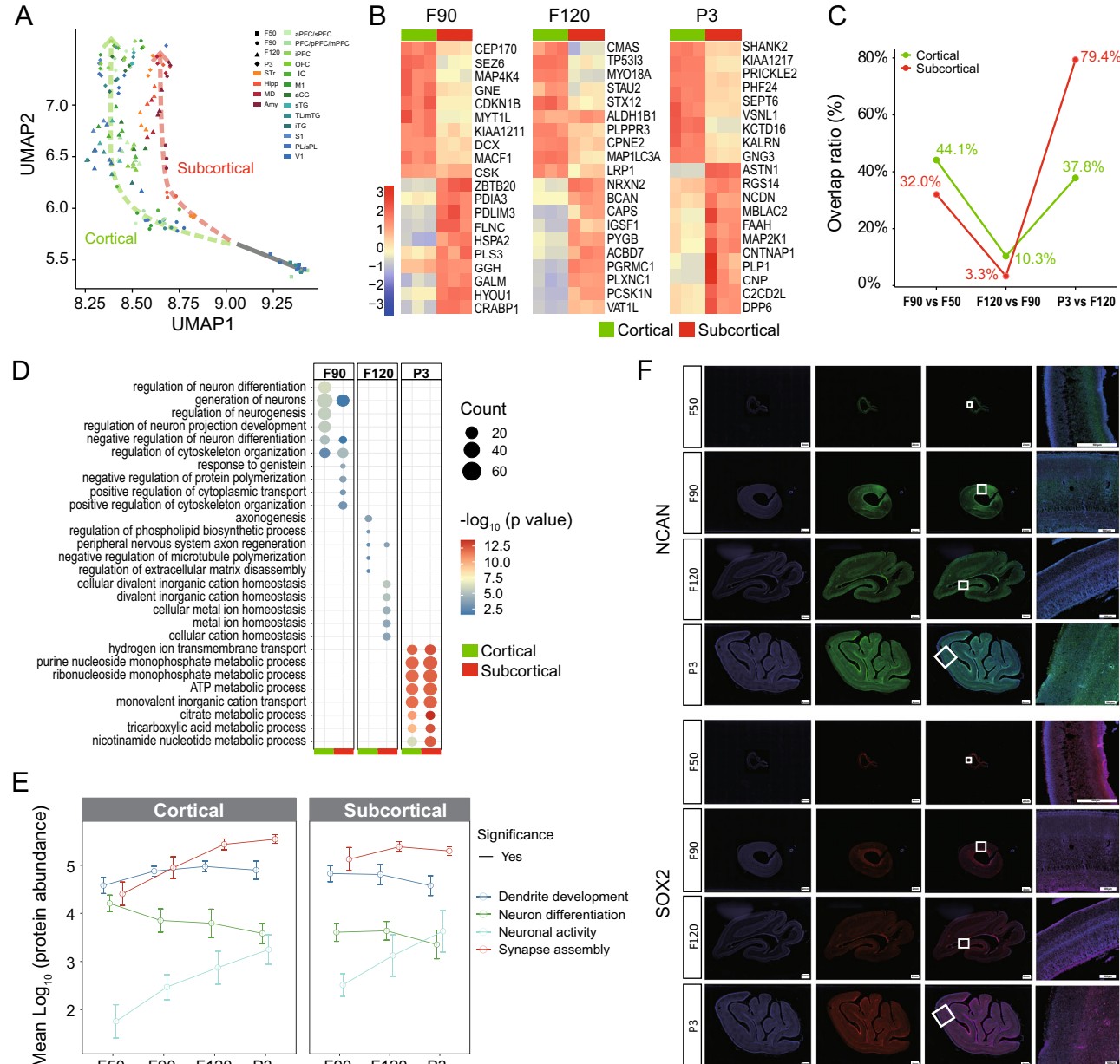

**Fig. 4 | Differences in proteomic profiles between cortical and subcortical regions. A** UMAP plot of brain samples showing the difference between cortical and subcortical regions. **B** Top 20 marker proteins in cortical and subcortical regions over the stages of F90, F120, and P3, respectively ($n = 3$). **C** Line chart showing the overlap rate of biological processes enriched by marker proteins between two adjacent stages. **D** Comparison of biological processes that become enriched at the indicated stages between cortical and subcortical regions. The newly enriched biological processes were identified by comparison of biological processes as in (**C**). The $p$-values were calculated by two-sided Fisher's exact test. **E** Line chart showing the trend over four stages of the protein related to dendrite development, neuron differentiation, neuronal activity, and synapse assembly, respectively. The cortical ($n = 12, 21, 39, 39$ subregions from three monkeys for

stages F50, F90, F120, P3, respectively) and subcortical ($n = 9, 12, 12$ regions from three monkeys for stages F90, F120, P3, respectively) regions in cerebrum were showed separately for comparison. The dot and error bar indicate the mean value and standard deviation, respectively. The solid line indicates that there is at least one significant difference between the two adjacent stages. The dashed line indicates no significant difference between any two adjacent stages. The $p$-values were calculated by two-sided Student's $t$-test. **F** Representative immunofluorescence images showing spatiotemporal changes in abundance of NCAN and SOX2. The brain tissues were stained for NCAN (green), SOX2 (red), and DAPI (blue), respectively. The scale bar is 2 mm (columns 1–4) or 500 μm (column 4) as indicated. Source data are provided as a Source Data file.

between cortical and subcortical regions during brain development. We identified marker proteins (upregulated proteins) of cortical and subcortical regions at stages F90, F120, and P3 (Fig. 4B and Supplementary Data 7). The result showed that cortical and subcortical regions presented stage-dependent marker proteins. The analysis of the overlap of biological processes enriched by marker proteins between two adjacent stages showed dynamic changes during fetal

brain development (Fig. 4C and Supplementary Data 8). In the cortical region, the overlap ratios of biological processes between adjacent stages were 44.1% (F90 vs F50), 10.3% (F120 vs F90), and 37.8% (P3 vs F120). In the subcortical region, the overlap ratios were 33.0%, 3.3%, and 79.4%, respectively, which showed a similar pattern to the cortical region. In the cortical region, the newly expressed proteins were involved in neurogenesis, axonogenesis, and multiple metabolic

processes in F90, F120, and P3 tissues, respectively (Fig. 4D). Similar processes also occurred in the subcortical regions at F90 and P3, except for ion homeostasis related processes at F120. It is worth noting that the overlap in enriched biological processes in F120 vs F90 (corresponding to 32 vs 24 weeks in human fetus) was the lowest in cortical and subcortical regions. Unlike other stages, the cortical and subcortical regions showed unmatched biological processes at F120. The results suggest that new biological processes might emerge to adapt to changes in brain structures and functions at the F120 stage, which corresponds to 32 weeks post-fertilization in humans.

We further analyzed the dynamic changes in average abundance and variation of proteins related to dendrite development, neuron differentiation, neuronal activity, and synapse assembly, which are important for brain development (Fig. 4E). The abundance of proteins associated with dendrite development and neuron differentiation decreased from F120 to P3 stage, whereas neuronal activity and synapse assembly proteins increased. Mfuzz analysis divided all proteins into four clusters with specific trends (Supplementary Fig. 5). It has been well known that SOX2 is a marker of neural stem cells, which play vital roles in pluripotency maintenance[31,32], and NCAN plays a role in synaptic plasticity, neuronal migration and/or formation of axonal fibers[33,34]. Mfuzz results showed that NCAN belonged to cluster 1 (increased trend) and SOX2 to cluster 2 (decreased trend) (Supplementary Fig. 5). Immunohistochemical staining showed the distributions of SOX2 and NCAN in the V1 region and confirmed the changes of the two proteins from our proteomic data (Fig. 4F).

## Dynamic proteomic changes among cortical regions and disease-related protein analysis

By UMAP analysis, we found the distributions of the four cortical regions were mixed together at each stage (Fig. 5A), which indicates that the overall protein profiling of cortical regions was similar at the same stage. However, the protein profiling of the cortical regions was separated along with developmental stages. Therefore, we compared every two adjacent stages by quantifying the number of new proteins and newly enriched biological processes (Fig. 5B). A new protein was defined as a protein that appeared at a given stage (LFQ intensity >0) but was not present in the previous adjacent stage (LFQ intensity =0). We observed that PFC presented more new proteins in the transition from F120 to P3 than the other regions (p-value < 0.01). During development, vital neurological processes, such as synaptogenesis and myelination, are highly activated in PFC[35]. We observed a higher rate of abundance change of synaptic proteins in the PFC at the transition from F50 to F90 compared to the other regions (p-value < 0.05), but not for all total proteins (Fig. 5C). Changes in the PFC are functionally related to neuropsychiatric diseases such as autism spectrum disorders (ASD)[36,37], major depressive disorder (MDD)[38], schizophrenia (SCZ)[39,40], and neurodegenerative diseases such as Alzheimer's disease (AD)[41,42] and Parkinson's disease (PD)[43,44]. We next sought to determine abundance changes in the reported proteins implicated in the five diseases[12]. We quantified the mean abundance of these proteins between PFC and V1 at the four stages (Fig. 5D). Proteins linked to neuropsychiatric diseases (ASD, MDD, and SCZ) were significantly changed in the PFC during fetal brain development. In contrast, the changes in the proteins linked to MDD and SCZ were not significant in V1. For proteins related to neurodegenerative diseases (AD and PD), the abundance showed a slight increase from F90 to P3 stages and changed significantly in V1 during fetal brain development. This result suggests that the pathogenesis of these diseases may be due to abnormal expression of the risk proteins at fetal stages.

## Cross-species proteomic comparison and differences between proteome and transcriptome

The macaque is believed to be an ideal non-human primate model for the research of the human brain. We therefore compared the cynomolgus monkey brain proteomes elucidated in this study with the recently reported human and mouse brain proteomes[18,26]. As the available human and mouse data did not cover all brain regions and did not exactly match the developmental time points of monkeys studied in our study, we only used the proteomic data from five macaque brain regions (STR, MD, HIPP, mPFC, and CB) of 3 days postnatal of monkeys to perform a comparison to the available proteomic data at the similar developmental stage of human[18] and mouse[26]. The three time points cross three species are all corresponding to the postnatal stage. Among the three species, a total of 3025 common ortholog proteins were used for comparison. We calculated the Pearson correlations for each 1:1 ortholog protein among the five regions under pairwise comparisons of Mouse vs Human, Monkey vs Mouse, and Monkey vs Human (Fig. 6A). More than 35% of orthologous proteins fall into the correlation interval of 0.75−1 when comparing monkey with human proteomes. The percentage was much higher compared to mouse and human (less than 10%). There was a median correlation of 0.011 for Mouse vs Human, but the correlation significantly increased (median correlation 0.605, KS p-value < 2.2e-16) for Monkey vs Human. Based on the Pearson correlations of 1:1 ortholog proteins, three classes of Human-specific (531), Primate-specific (591), and Conserved (457) proteins were defined. The human-specific proteins were mainly enriched in Golgi vesicle transport, macroautophagy, and so on. The primate-specific proteins were mainly enriched in neuron development, anterograde trans-synaptic signaling, and so on. The conserved proteins were mainly enriched in dendritic spine development, RNA splicing, and so on (Fig. 6B). In addition, we also analyzed the correlations for each protein in the top 10 abundant protein families among the five regions under the three pairwise comparisons (Fig. 6C and Supplementary Fig. 6). Each protein family presented proteins with both positive and negative correlations in either of the three pairwise comparisons. The monkey vs human comparison showed more proteins falling into the 0.75−1 correlation interval than those of the mouse vs human and monkey vs mouse. Although the overall similarity was high, we found the protein abundance of several specific members was distinguished between humans and monkeys. This result suggested the protein family was regulated in a species-specific manner. For example, in the tubulin family, TUBG1 and TUBA4A were highly correlated between monkey and human, while TUBB4A and TUBB6 were negatively correlated. In the protein kinase superfamily, several proteins such as TAOK2, PTK2, MAST1, MAP2K4, MAP2K1 have a correlation of 1 between monkey and human, while several proteins were negatively correlated. The species specificity and increasing similarity between macaque to human highlight the usefulness of this data to understand human brain development since the human fetal brain proteome is unavailable so far.

In order to reveal the differences between RNA and protein levels and the characteristics of post-transcriptional regulation, we compared the changes in our proteomic data from prenatal (F120) and postnatal (P3) brains to the previously published macaque transcriptomic data from prenatal (F110) and postnatal (P2) stages[12]. Based on the FC of RNA and protein levels, all genes were divided into six types (Fig. 6D). Type 1−3 genes accounted for more than 90% of all genes. Type1 genes were defined as the genes with small changes in both RNA and protein levels (|FC| < 2). Both type2 and type3 genes showed a large change between RNA and protein levels. The percentage of type 1−3 genes in FL was 78.7%, 15.6%, and 1.4%, respectively. The results highlighted a certain proportion (>15%) of genes with mRNA/protein discrepancies. A similar pattern was found in other cortical regions, as well as in subcortical regions and the CB (Supplementary Fig. 7). Type1 genes were predominantly enriched in the processes related to axonogenesis and neuronal projections (Supplementary Fig. 8). Type2 genes were predominantly enriched in transcription related processes. Type3 genes were predominantly enriched in phagocytosis, cell adhesion and regulation of cell migration, etc. We

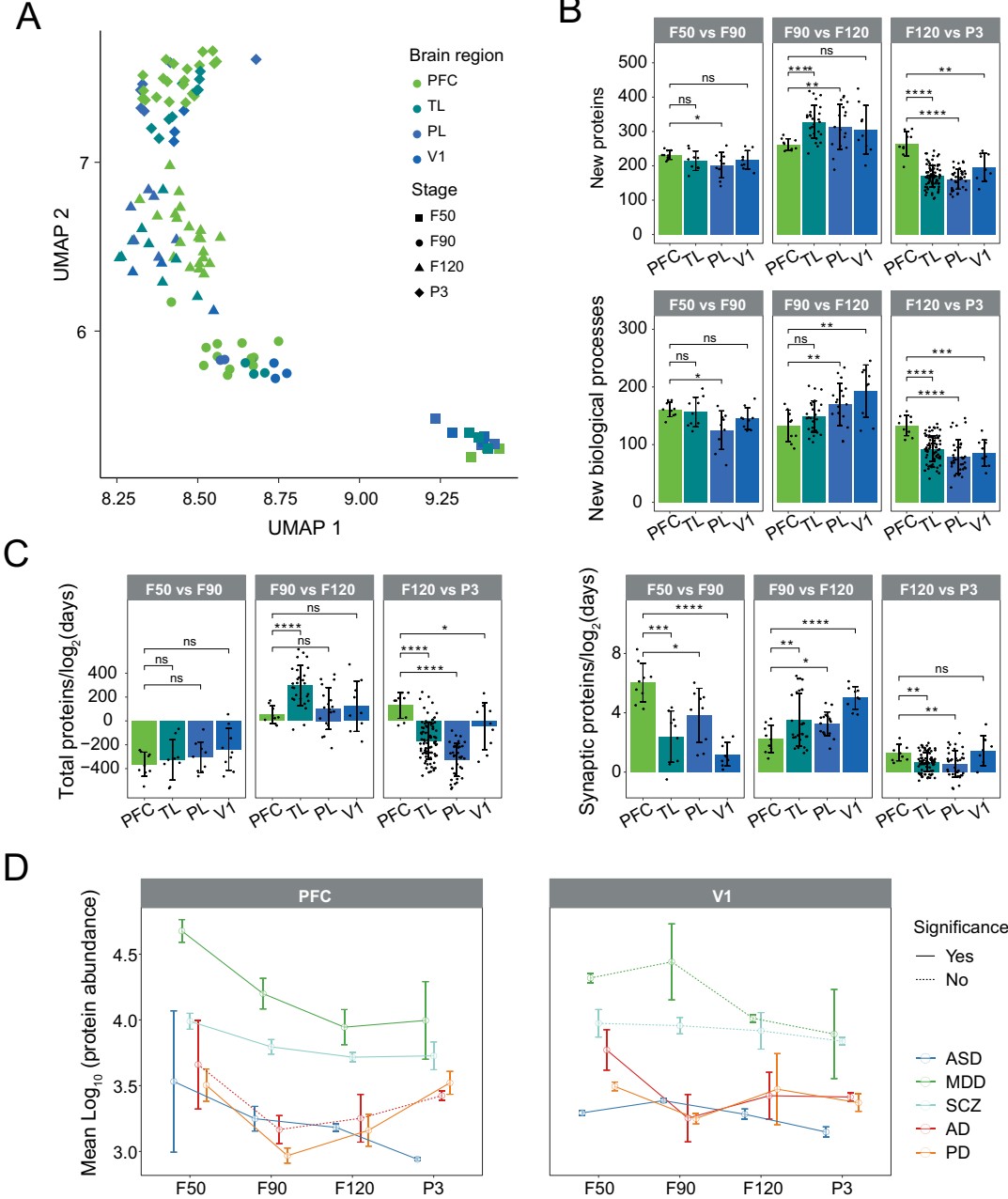

**Fig. 5 | Dynamic proteomic changes in cortical regions. A** UMAP plot showing the distribution of brain samples from cortical regions. **B** The number of the new proteins (top panel) and newly enriched biological processes (bottom panel) in the cortical regions (PFC, TL, PL, and V1) when comparing the adjacent stages. **C** The rate of abundance changes of total proteins (left panel) and synaptic proteins (right panel) between two adjacent stages. In panels (**B**) and (**C**), for the F50 vs F90 comparison, $n = 9$ subregions from three monkeys for PFC, TL, PL, and V1. For the F90 vs F120 comparison, $n = 9, 27, 18, 9$ subregions from three monkeys for PFC, TL, PL, and V1, respectively. For the F120 vs P3 comparison, $n = 9, 81, 36, 9$ from three monkeys for PFC, TL, PL, and V1, respectively. ns: $p > 0.05$, *: $p < 0.05$, **: $p < 0.01$,

***: $p < 0.001$, or ****: $p < 0.0001$. Data are presented as means ± SD. **D** Line chart showing the trend over the four stages of the proteins related to AD, ASD, MDD, PD, and SCZ, respectively ($n = 3$). The dot and error bar indicate the mean value and standard deviation, respectively. The solid line indicates that there is at least one significant difference between the two adjacent stages. The dashed line indicates no significant difference between any two adjacent stages. The $p$-values were calculated by two-sided Student's $t$-test (**B–D**). PFC prefrontal cortex, TL temporal lobe, PL parietal lobe, V1 primary visual cortex, AD Alzheimer's disease, ASD autism spectrum disorder, MDD major depressive disorder, PD Parkinson's disease, SCZ schizophrenia. Source data are provided as a Source Data file.

observed that the neuronal projection process was a conserved process that was enriched by Type1 genes in all brain regions (Supplementary Figs. 8 and 9 and Supplementary Data 9). Type 4–6 genes accounted for less than 5% of all genes. Type4 genes showed consistent direction and fold change >2 at RNA level or at protein level. Type5 genes showed opposite direction between RNA level and protein level. Type6 genes showed consistent direction and fold change >2 at both RNA level and protein level. Type4 genes were mainly enriched in the

angiotensin-related processes, lipid phosphorylation, coagulation, hemostasis, etc. Type5 genes were mainly enriched in the cytoskeleton-related processes. Type6 genes were mainly enriched in protein localization to kinetochore, microvillus assembly, attachment of spindle microtubules to kinetochore, long-term synaptic depression, etc. In other brain regions, these six types of genes showed region-specific characteristic and are therefore enriched for different biological functions and processes (Supplementary Figs. 10 and 11 and

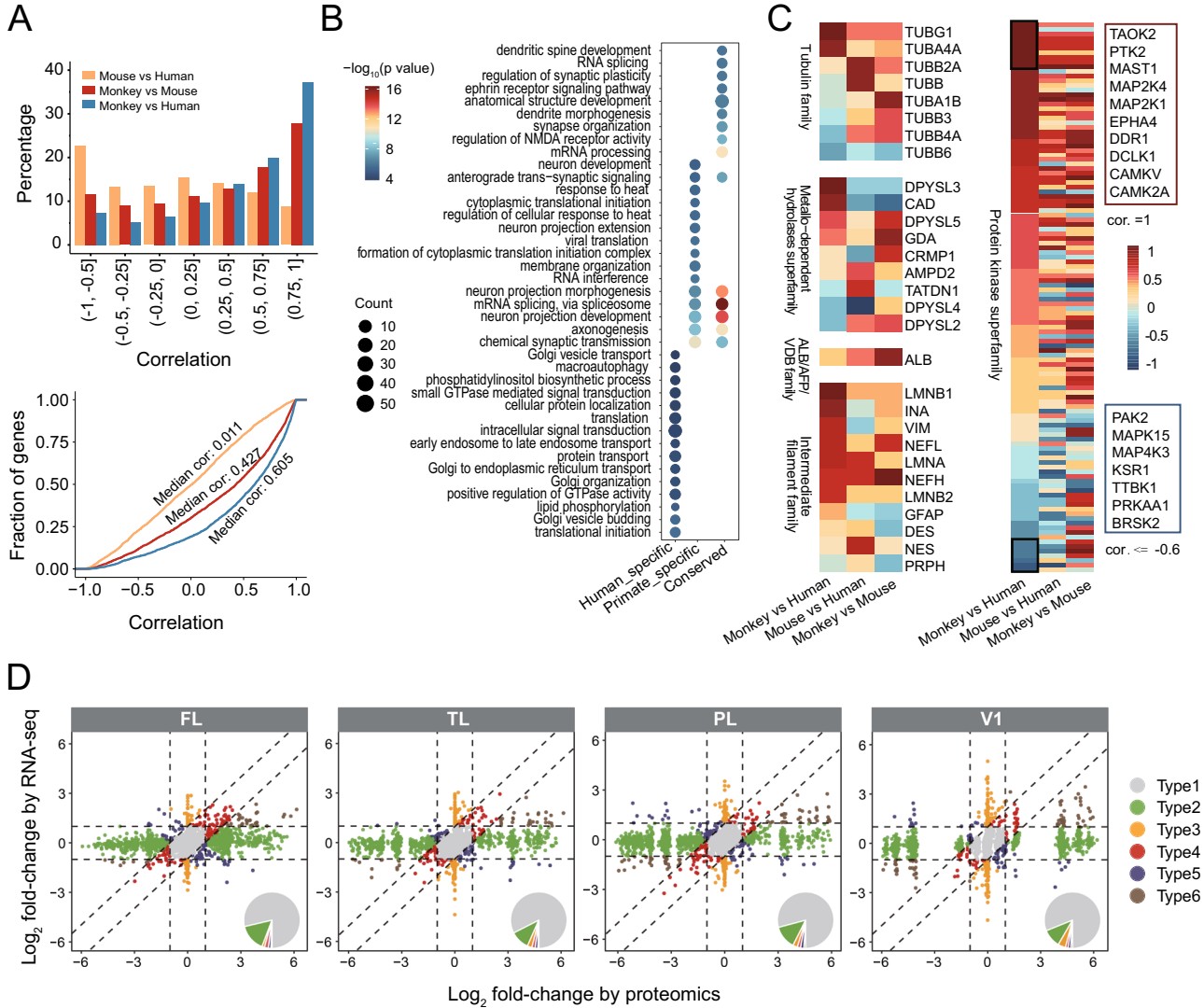

**Fig. 6 | Comparison of the human, monkey, and mouse brain proteomes.**
**A** Pairwise Pearson correlations of brain proteomic dataset of Mouse, Monkey, and Human (PFC, STr, MD, Hipp, and CBC). A total of 3025 common ortholog proteins among the three species were used. Bar chart and line chart showing the percentage and cumulative frequency of Pearson correlations for each 1:1 ortholog protein ($n = 3025$) under the three pairwise comparisons. **B** The enriched biological processes by the three classes of Human-specific (531), Primate-specific (591), and Conserved (457) proteins. **C** Heatmap showing Pearson correlations among the five brain regions for each protein in the top five abundant protein families under the three pairwise comparisons. **D** The scatter plot showing the fold changes (FC) between prenatal and postnatal for the RNA-seq and proteomics in the four cortical regions (FL, TL, PL, V1). Based on the FC of RNA and protein levels, all genes can be divided into six types as defined in "Methods". The six types of genes were indicated by different colors. Inset pie charts illustrate the relative percentages of the six types of genes. mPFC Medial prefrontal cortex, TL temporal lobe, PL parietal lobe, V1 primary visual cortex, STR striatum, MD mediodorsal nucleus of the thalamus, HIPP hippocampus, CB cerebellum. Source data are provided as a Source Data file.

Supplementary Data 9). These results emphasize the importance of proteomic studies of brain development and is an important complement to previous transcriptomic studies.

## Discussion

This study presents high-resolution proteomic data from multiple brain regions spanning the fetal to neonatal developmental stages in a non-human primate. The resulting resource is beneficial for understanding both brain function during normal development and mechanisms of dysfunction and disease in humans. This is because there is currently a lack of data at these developmental stages in healthy humans due to ethical issues and difficulties in sampling at these time points.

We found that the abundance of the protein families changed differently during fetal neurodevelopment, for example, the tubulin family, ALB/AFP/VDB family, and protein kinase superfamily. The proteins of the tubulin family are subunits of microtubules, and different tubulins isoforms may be required for specific microtubule functions, which are crucial for cortical development, including neuronal proliferation, migration, and cortical laminar organization. Mutations of tubulin proteins have been identified in individuals with a range of tubulin-related diseases, such as lissencephaly, schizencephaly, and microlissencephaly[45]. In this study, we found that the abundances of the tubulin family in cortical areas showed ascending trend from F50 to F90, which suggests this period may be critical for the formation of the cerebral cortex. In contrast, a decreasing trend from F90 to P3 was observed, which indicates a decreasing proportion of neurons and an increasing proportion of non-neuronal cells during this period. As a member of the ALB/AFP/VDB family, the AFP (alphafetoprotein) is abundant in the brain during embryonic development and inhibits neurite growth and neuron migration by restraining the neurotrophic effect of oleic acid[46]. We found that the abundance of

AFP was continuously decreased from F50 to P3, suggesting that AFP may be an inverse indicator of neurodevelopment. As a member of the protein kinases family, TAOK2 is essential for the formation and stability of dendritic spines as well as axon elongation in cortical neurons[47]. Loss of TAOK2 activity causes autism-related neurodevelopmental and cognitive abnormality[48]. In this study, TAOK2, with a correlation of 1 between monkey and human, showed a continuously increasing trend from F50 to P3, which indicates that TAOK2 plays an important role in the maintenance of normal neurodevelopment.

Previous transcriptomic studies demonstrated an overall cup-shaped developmental dynamic in fetus brain development both in human and rhesus monkey[12,49]. The bottom of cup-shaped transcriptomic variation coincides with late fetal and neonatal stages. Similarly, in our study, the pattern of DEPs between brain regions did not continuously increase over time and also exhibited an overall "V" shaped pattern. Our proteomic data revealed that the lowest inter-regional variation was at F90 in CB and F120 in CTX and sCTX, respectively, which was earlier than the finding from transcriptomic data. This result suggests the development of the brain is an intrinsic program dominated before F120 and shifts to an environment and heredity interacting mode after birth. Such brain developmental characteristics observed in humans and monkeys imply this transition process is a conserved feature of primate neurodevelopment. Notably, the data showed that there was a dramatic reduction in inter-regional differences in CB at F90, which may be due to technical reasons. At F90, one CB sample was lost and three technical replicates were performed on two CB samples instead. The high similarity of the results of the three LC/MS tests resulted in a low value of inter-regional differences in CB at F90. Based on the trend shown in CB, it is reasonable to speculate that the transition time of CB is also at the F120 stage.

In the comparison of CB and other brain regions, the proteomic data revealed that CB had a distinct developmental pattern and showed increasing Euclidean distances along with brain development. The distance between the CB and V1 did not increase with development. We identified the upregulated proteins as CB marker proteins at each stage. Enrichment analysis of the marker proteins emphasizes distinct biological processes and pathways at different developmental stages. For example, the enrichment of mRNA processing-related processes was observed both at F120 and P3 stages, which suggests that trends in the CB at late fetal stages continue after birth. The result was consistent with information from the postnatal human brain[18], suggesting a specialization of CB during the perinatal period in primates. The GRID2IP, PCP2, ZIC1, and GABRA6 were observed to highly express at F120 and P3 stages. These proteins were known to be related to the maturation of Purkinje cells and granule cells[28,29], respectively. The above results together with the findings that CB had an earlier reflection point of inter-regional differences suggest that neural differentiation in the CB might start at F90, which is earlier than neural differentiation in the cerebrum.

In the context of the enormous cellular diversity and specialization in the brain, researchers have been trying to unravel how the proteome differs between functionally specialized regions. Previous proteomic studies have found that temporal variation is much smaller than regional variation in postnatal human brain[19]. However, our data show that the temporal variations in proteomes are larger than the regional variations in brain development from fetal to neonatal stages. The prominent temporal variations at these stages may due be to the dramatic increase of cell diversity following fetal brain development. However, after birth, most cells have completed terminal differentiation and acquired specific functions in different brain regions. The rapid formation and pruning of synapses and neural circuit trimming after birth require dramatic changes in proteins, which may drive the significant differences between cell types and functions in brain regions.

Processes including dendrite development, neuronal differentiation, neuronal activity, and synapse assembly become prominently enriched during late fetal brain development. Previous transcriptomic studies showed that the expression trajectories of these processes in cortical regions were synchronous[12,49]. Our study, at the protein level, revealed similar synchronous trends in cortical regions, as well as in subcortical regions. It has been reported that the mid- and late fetal stages are a key developmental period in humans and that these stages are also associated with the etiology of neuropsychiatric and neurodegenerative disease[12,50,51]. Our proteomic data demonstrate that protein changes related to neuropsychiatric and neurodegenerative diseases showed opposite trajectories both in PFC and V1. Proteins of risk genes linked to neuropsychiatric diseases were significantly changed in the PFC during fetal brain development. In contrast, the changes were not significant in V1. This finding suggests that the functional execution of the PFC may be more sensitive to neuropsychiatric diseases than V1. To date, previous studies barely reported the relationship between the primary visual cortex (V1) and neurodegenerative diseases. Our data revealed the proteins of risk genes associated with AD and PD changed significantly in V1 during fetal brain development, which implies further investigation on the roles of V1 in the pathogenesis of degenerative diseases deserves more attention. Confirmation of this hypothesis in future work may allow early detection and treatment of these conditions.

Our result revealed a certain proportion (>15%) of genes with mRNA/protein discrepancies. There are multiple reasons that could explain the discrepancy between RNA and protein expression[52]. The mRNA may be modified by methylation (e.g., m6a) and their stability and translational efficiency are dependent on these modifications. Protein stability also contributes to the discrepancy between RNA and protein expression. Some protein has a long half-life while others are immediately destroyed for proper function. There is alternative splicing in post-transcription. Some splice variants are transcribed but not translated into protein. In addition, the miRNA and lncRNAs are regulatory factors that can control translation efficiencies.

This study provides comprehensive proteomic data across development from fetal to neonatal stages in cynomolgus macaque which has many potential applications. For example, it can be used as an in vivo reference of fetal brain development that supports future research into abnormal brain development and diseases. The present resource may also aid the development and application of brain organoids in vitro. Integration of this resource with previously reported transcriptomic and epigenomic data could provide insights into brain development. This resource is publicly accessible via https://db.cngb.org/search/project/CNP0002788/. In summary, our study characterizes the dynamic pattern of protein abundance during fetal brain development in primates and provides a proteomic data resource that has profound implications for the neuroscience community.

## Methods

### Animals
Tissue samples were obtained from fetal and early postnatal cynomolgus monkeys (*Macaca fascicularis*) resulting from natural conception. Offspring were from healthy adult female (maternal) and male (paternal) monkeys aged from 7 to 8 years. The female monkeys were observed for at least two normal menstrual cycles before being allowed to mate with male monkeys. The determination of fertilization and developmental stage referred to our previous study[53]. Briefly, the levels of Estradiol (E2) and progesterone (P4) in venous blood were assayed using a chemiluminescent immunoassay. The levels of E2 and P4 in venous blood were continuously measured and the day of ovulation and fertilization was set as the day after the peak value of E2 was detected. The pregnancy was confirmed by the low level of E2, increased level of P4, and ultrasonography. Tissue from offspring at 50, 90, and 120 days post-fertilization and 3 days post-birth were

labeled F50, F90, F120, and P3, respectively. Fetuses (three for each stage) were obtained by caesarean section, and postnatal monkeys (three animals) were born naturally. The parents of nine fetuses and three postnatal monkeys were individually raised in an animal room with humidity at 40–70%, temperature at 18–26 °C, and a 12/12 light–dark cycle.

The experimental plan was approved by the Institutional Animal Care and Use Committee of Kunming University of Science and Technology in advance (approval number: LPBR201801020). All procedures involving animals were performed in accordance with the Guide for the Care and Use of Laboratory Animals (the 8th edition, NIH). All cynomolgus monkeys were housed and raised at the facility of the State Key Laboratory of Primate Biomedical Research.

### Brain tissue dissection
Postnatal animals were euthanized using pentobarbital sodium (100 mg/kg, i.v.). For all specimens (fetus and postnatal), the brain tissue was collected using a surgical blade. Brain regions of the left hemisphere were systematically dissected using gyri and sulci as landmarks (Fig. 1A). Histological atlases of the cynomolgus monkey brain at the Brain Maps site (http://www.brainmaps.org) were used to define the location of brain regions. For the monkey, five regions (PFC, TL, PL, V1, and CB) in F50, 11 regions (aPFC, pPFC, IC, M1, TL, PL, V1, CB, STr, Hipp, and Amy) at F90, and 18 regions (sPFG, mPFG, iPFG, OFC, IC, M1, aCG, sTG, mTG, iTG, S1, sPL, V1, CB, STr, Hipp, MD, and Amy) at the F120 and P3 stages were carefully dissected by one experienced neuroanatomist. Given that brain regions vary in size, a tissue volume of 0.5–2 mm³ was collected for different brain regions. The dissected tissue was frozen in liquid nitrogen immediately and then stored at −80 °C until processing. The right hemisphere of each specimen was fixed in 4% paraformaldehyde (0.01 M PBS) for histological examination. The whole dissection process was completed in 20 min. The information on the samples from the fetuses and postnatal monkeys, including animal code, developmental stage, sex, and brain regions, was listed in Supplementary Table 1. Notably, three biological repeats were performed on all brain regions of the four stages except CB at the F90 stage. As we lost one CB sample at the F90 stage, three technical replicates were performed on two CB samples at the F90 stage instead.

### Sample preparation
Samples were ground in liquid nitrogen into powder and then transferred to a 5-mL centrifuge tube. Four volumes of lysis buffer (8 M urea, 1% Protease Inhibitor Cocktail) were added to the cell powder, followed by sonication three times on ice using a high-intensity ultrasonic processor (Scientz). (Note: For PTM experiments, inhibitors were also added to the lysis buffer, e.g., 3 μM TSA and 50 mM NAM for acetylation.) The remaining debris was removed by centrifugation at 12,000×g at 4 °C for 10 min. Finally, the supernatant was collected and the protein concentration was determined with the BCA kit (71285-M, Millipore) according to the manufacturer's instructions.

For digestion, the protein solution was reduced with 5 mM dithiothreitol for 30 min at 56 °C and alkylated with 11 mM iodoacetamide for 15 min at room temperature in darkness. The protein sample was then diluted by adding 100 mM TEAB to a urea concentration of less than 2 M. Finally, trypsin was added at 1:50 trypsin-to-protein mass ratio for the first digestion overnight and 1:100 trypsin-to-protein mass ratio for a second 4-h digestion.

### Mass spectrometry analysis (LC-MS/MS)
The tryptic peptides were dissolved in 0.1% formic acid (solvent A) and then directly loaded onto a homemade reversed-phase analytical column (15 cm length, 75 μmi.d.). The gradient comprised an increase from 6 to 23% solvent B (0.1% formic acid in 98% acetonitrile) over 26 min, 23 to 35% in 8 min, increasing to 80% in 3 min, then holding at 80% for the last 3 min, all at a constant flow rate of 400 nL/min on an EASY-nLC 1000 UPLC system. The peptides were subjected to a NanoSpray Ionization source followed by tandem mass spectrometry (MS/MS) in Q ExactiveTM Plus (ThermoFisher) coupled online to the UPLC. The electrospray voltage applied was 2.0 kV. The m/z scan range was 350 to 1800 for a full scan and intact peptides were detected in the Orbitrap at a resolution of 70,000. Peptides were then selected for MS/MS using NCE setting 28 and the fragments were detected in the Orbitrap at a resolution of 17,500. A data-dependent procedure alternated between one MS scan followed by 20 MS/MS scans with 15.0 s dynamic exclusion. Automatic gain control (AGC) was set at 5E4 and the fixed first mass was set as 100 m/z. The resulting MS/MS data were processed using the Maxquant search engine (v.1.5.2.8). Tandem mass spectra were searched against the UniProt protein database (Macaca_fascicularis_9541_PR_20190905) concatenated with a reverse decoy database. Trypsin/P was specified as the cleavage enzyme allowing up to four missing cleavages. The mass tolerance for precursor ions was set as 20 ppm in the First search and 5 ppm in the Main search, and the mass tolerance for fragment ions was set as 0.02 Da. Carbamidomethyl on Cys was specified as a fixed modification and acetylation modification and oxidation on Met were specified as variable modifications. The collection of proteomic data was performed by a professional company for mass spectrometry analysis (Jingjie PTM BioLab Co. Ltd, Hangzhou, CN)

### Protein identification and quantification
In this study, we used the method of label-free quantification (LFQ) to determine the relative abundance of proteins in our samples. Peptide was identified with a posterior error probability (PEP) score <0.05 and a minimum score for modified peptides >40. False discovery rate (FDR) thresholds for protein and peptide were specified at 1%. The minimum peptide length was set at 7. All other parameters in MaxQuant were set to default values. A total of 33314103 secondary spectra were obtained from the mass spectrometry analysis. After searching the protein theoretical data, the number of usable secondary spectra was 6760839, which is 20.3% of the spectrum utilization rate. A total of 120,730 peptides were identified by spectral analysis, of which 97,173 were unique peptides that match a single protein. The other peptides match multiple proteins, which are called "razor" peptides. The "razor" peptides are assigned to the protein group with the most unique peptides. Then the relative intensity of these peptides was quantified as the area of the peak that was extracted from the primary mass spectrometry (MS1). Finally, MaxQuant employs the MaxLFQ algorithm for label-free quantitation (LFQ)[54]. The intensities of individual proteins will be corrected for quantitative information based on the peptides identified across samples, and the LFQ intensity will be calculated to minimize inter-sample errors caused by handling, loading, pre-sorting, instrumentation, etc., to allow quantitative comparisons of the same proteins across samples. The LFQ intensity >0 means the protein is detected in this sample. Raw results for the identification and quantification of all peptides and proteins are shown in Supplementary Data 10. LFQ intensities of proteins were used for downstream analysis and statistics.

### Downstream analysis and statistics
All downstream analyses and statistics were performed using R/Bioconductor. The PCA analysis was performed using the function "prcomp" in the R. The UMAP analysis was performed using the R package umap with default parameters except for "n_neighbors=30".

The marker proteins for each of the identity classes were identified using the function "FindAllMarkers" in the Seurat package with default parameters except for test.use = "roc". Protein with AUC > 0.7 and fold change >1 were considered marker proteins. The protein lists involved in dendrite development, neuron differentiation, neuronal activity, and synapse assembly, were also based on this reference. Functional enrichment analysis was performed using the function

"enricher" in the R package clusterProfiler[55]. The R package ggplot2 was used for data visualization. Unless explicitly stated, all statistical tests are performed using the Student's t-test with Bonferroni correction. The difference was considered statistically significant when the adjusted p-value was <0.05.

Specifically, in Fig. 1, we examined protein abundance, the number of proteins detected, the number of differentially expressed proteins at different times and stages, and the proportion of protein subcellular localization and protein families. The annotation of protein subcellular localization was based on The Human Protein Atlas (https://v18.proteinatlas.org/)[20]. The annotation of the protein family was based on the superfamily subset in InterPro[56] (https://www.ebi.ac.uk/interpro/). Protein abundance is quantified as LFQ intensity for each protein. The distribution of log2 transformed protein abundance was shown using a boxplot (Fig. 1B). The number of detected proteins in each sample was shown as a scatterplot (Fig. 1C). In the same stage, differentially expressed proteins (DEP) in a specific region are identified by comparing the differences between the specific region and all other regions using t-test with Bonferroni correction. The protein with adjusted p-value < 0.05 was considered as DEP (Fig. 1D). Calculating the percentage of subcellular localization was described as follows. For each sample, the percentage of a particular subcellular localization is obtained by adding the LFQ intensities of all proteins belonging to a particular cellular localization and then dividing by the LFQ intensities of all proteins in the sample. Based on the percentages of subcellular localizations in each sample, we can calculate the average percentages in different regions or at different stages (Fig. 1E). The t-test is used to test for significance when comparing percentages at different stages in a given region. Using the same logic, the percentages of the protein families were also calculated and compared (Fig. 1E).

Next, we explored the dynamic proteomic changes across developmental stages and brain regions (Fig. 2). PCA and heatmap were used to show the distribution and correlation of samples, respectively. Pearson correlation coefficient and the p-value were calculated between stages using cor and cor.test in R language (Fig. 2B). DEP among stages or regions in Fig. 2C was calculated as Fig. 1D. All identified DEPs were summed by stage and region, respectively. The results for all regions (excluding the CB region), cortical areas subcortical areas are shown by the Venn diagram. Next, we grouped samples by regions (CB, cortical areas, subcortical areas) and calculated the interregional differences at each developmental stage (Fig. 2D). The interregional differences were calculated as the average of absolute difference of log2 transformed LFQ intensities of paired proteins from all pairs of samples in the same grouped area. The significance of differences between adjacent time points was calculated using t-test. We grouped the samples of all regions (CB excluded) at each stage, and the number of samples at the four stages for statistical comparison were 12 (F50), 30 (F90), 51 (F90), and 51(P3), respectively. The marker proteins in each stage were identified using the function "FindAllMarkers" as described above. The normalized LFQ intensities of the top 10 marker proteins sorted by AUC value for each stage were shown as a heatmap (Fig. 2E).

We examined the differences of CB to other regions at each stage by PCA (Fig. 3A) plot and Euclidean distance calculation (Fig. 3B). After identification of CB marker proteins using the function "FindAllMarkers" as described above, the top 20 marker proteins sorted by AUC value for each stage were shown as a heatmap (Fig. 3C). The biological process involved these CB marker proteins were enriched by using the R package clusterProfiler (Supplementary Fig. 3).

The dynamic proteomic changes and differences between cortical and subcortical were investigated by the UMAP plot (Fig. 4A), the marker proteins (Fig. 4B), and the important brain development-related processes (Supplementary Data 8). The protein lists involved in dendrite development, neuron differentiation,

neuronal activity, and synapse assembly, were obtained the study of Li et al.[12]. Here we focus on describing the calculation of the overlap rate of biological processes in Fig. 4C, which refers to the previous study[2]. For cortical and subcortical, the marker protein was first calculated separately for the two regions at each stage. Then the biological processes were enriched based on the marker protein at each stage. When the biological processes for each stage were obtained, the similarity of the biological processes between two adjacent developmental stages was calculated. The newly enriched biological processes at the indicated stages were identified by comparison of the biological processes at the pre-stage of this stage. The newly enriched biological processes in cortical and subcortical regions were calculated and shown in Fig. 4D, respectively.

We also investigated changes in the relative abundance of new proteins, synaptic proteins, and disease-related proteins in different cortical regions with developmental stages (Fig. 5). A new protein was defined as a protein that appeared at a given stage (LFQ intensity >0) but was not present in the previous adjacent stage (LFQ intensity =0) (Fig. 5B). A new protein was also satisfied with the condition (p-value < 0.05 and fold change >1). Emerging proteins were found by comparing the pre- and post-phase samples (two by two) and then filtering for significantly upregulated proteins between pre- and post-phase. New biological processes were enriched by these new proteins. The temporal rates of change of total and synaptic proteins were obtained by calculating the difference between the two phases before and after divided by the number of days elapsed (Fig. 5C). For the PFC and V1, the relative abundance of each of the disease-associated proteins (AD, ASD, MDD, PD, SCZ) in each sample was summed up at each stage. The protein lists related to Alzheimer's disease (AD), autism spectrum disorder (ASD), major depressive disorder (MDD), Parkinson's disease (PD), and schizophrenia (SCZ) were obtained from the study by Li, M. et al.[12]. The summed abundances were then averaged to give the average abundance of that disease-associated protein at that stage (Fig. 5D). The significance of changes in two adjacent stages was tested using a t-test.

Finally, we performed comparison of a cross-species proteomics and differences in changes of proteomic and transcriptomic in monkeys before and after birth. To compare the monkey brain proteomes with human and mouse, human brain proteomes in five regions STR, MD, HIPP, mPFC, and CB were acquired from Carlyle et al.[18]. Mouse brain proteomes were acquired from Kirti et al.[26]. There is no available data that exactly match the developmental time points of mouse, monkey, and human. Therefore, we only used the proteomic data of 3 days postnatal of monkey to perform a comparison with the proteomic data at a similar developmental stage of human (86 days after birth, from Carlyle et al.[18]) and mouse (63 days after birth, from Kirti et al.[26]). The three time points cross three species are all corresponding to the postnatal stage. A total of 3025 common ortholog proteins among the three species were used for this analysis. We calculated the percentage and cumulative frequency of Pearson correlations for each 1:1 ortholog protein (n = 3025) under pairwise comparisons of mouse vs human, Monkey vs Mouse, and Monkey vs Human, respectively (Fig. 6A). Based on the Pearson correlations of 1:1 ortholog proteins, three classes of 'Human specific', 'Primate specific', 'Conserved' proteins were defined. Human-specific proteins are defined as a concatenation of proteins correlating between −1 and −0.5 in monkey-human and mouse-human comparisons. Primate-specific proteins are proteins with a correlation of 0.75 to 1 in the monkey-human comparison minus proteins with a correlation of 0.75 to 1 in the mouse-human and monkey-mouse comparisons. Conserved proteins are the intersections of proteins between 0.75 and 1 in monkey-human and monkey-mouse comparisons. Then three classes of proteins were performed enrichment analysis of

biological processes using the function "enricher" in the R package clusterProfiler. The results of the enrichment analysis are shown in (Fig. 6B). Furthermore, we also calculated the Pearson correlations among the five regions for each protein in the top 10 abundant protein families under the three pairwise comparisons (Fig. 6C and Supplementary Fig. 6). For comparison of transcriptome and proteome, transcriptome data was obtained from Li, M. et al.[12]. We compared the proteomic data from our prenatal (F120) and postnatal (P3) monkeys to the previously published monkey transcriptomic data[12] from prenatal (110 days post-fertilization) and postnatal (2 days after birth) stages. We calculated the fold changes (FC) between prenatal and postnatal for the transcriptome and proteome, respectively, in different brain regions (FL, TL, PL, V1, STr, Hipp, MD, Amy, CB) (Fig. 6D and Supplementary Fig. 7). Based on FC of RNA and protein levels, all genes can be divided into six types. Type1 genes were defined as the genes with small changes at both RNA level and protein level ($|FC| < 2$). Type2 genes were defined as the genes with higher change at protein level ($|FC| > 2$) than at RNA level ($|FC| < 2$). Type3 genes were defined as the genes with higher change at RNA level ($|FC| > 2$) than at protein level ($|FC| < 2$). Type4 genes were defined as the genes with consistent direction and fold change >2 at RNA level or at protein level. Type5 genes were defined as the genes with opposite direction between RNA level and protein level. Type6 genes were defined as the genes with consistent direction and fold change >2 at both RNA level and protein level. The proportions of the six gene types are shown in a pie chart. The six types of genes were performed enrichment analysis using the function "enricher" in the R package clusterProfiler. The results of the enrichment analysis are shown in Supplementary Figs. 8–11 and Supplementary Data 9.

### Immunohistological assay

Tissues for immunohistological analysis were sectioned using a Leica freezing microtome (CM 1860 UV) with a thickness of 15 μm after gradual dehydration with 15, 20, and 30% sucrose solutions.

The frozen sections were washed with PBS and blocked with 1% BSA+0.3% Triton X-100 in PBS for 2 h at room temperature. Subsequently, the tissues were incubated with primary antibodies overnight at 4 °C. The primary antibodies used were rabbit anti-SOX2 (AB5603, Millipore, 1:500) and mouse anti-Neurocan/NCAN (ab31979, Abcam, 1:500). The sections were further incubated with goat anti-mouse (A11029, Invitrogen, 1:500) with Alexa Fluor 488 or goat anti-rabbit IgG (A11012, Invitrogen, 1:500) with Alexa Fluor 594 for 2 h. The slices were sealed with the fluorescent-shield mounting agent with DAPI (ab104139, Abcam, 1:500) for 5 min. Immunofluorescence images were observed and photographed using a Leica SP8 confocal microscope.

### Reporting summary

Further information on research design is available in the Nature Portfolio Reporting Summary linked to this article.

## Data availability

The mass spectrometry proteomics raw data generated in this study have been deposited in the China National GeneBank DataBase (CNGBdb) under accession code CNP0002788 and are publicly accessible at https://db.cngb.org. We provided an interactive web page (https://cpbra.cn/menu_details/cyno/cBrainDev). All processed data have also been made publicly available via Zenodo (https://doi.org/10.5281/zenodo.7979998)[57]. Source data are provided with this paper.

## Code availability

All processed data and analysis scripts have been made publicly available via Zenodo (https://doi.org/10.5281/zenodo.7979998)[57].

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

## Acknowledgements

This study was supported by grants from the National Key Research and Development Program of China of 2018YFA0801400 (W.S.) and 2018YFA0108503 (J.W.), the National Natural Science Foundation of China of 32260197 (J.W.) and 32160153 (S.D.), the Natural Science Foundation of Yunnan Province of 202001BC070001 (W.J.), 202102AA100053 (W.J.), 202202AG050018 (W.S.), 201101AT070278 (J.W.), 202105AD160008 (S.D.), 202207AA110003 (S.D.) and 202105AC160041 (Z.W.).

## Author contributions

J.W., S.L., Y.Y., T.H., X.L., Y.D., and Z.W. conducted biological experiments and data collection. S.D., P.Y., and R.Z. performed proteomic analysis. J.W. and W.S. conceived and designed the experiments and analyses. J.W., S.D., Z.W., and W.S. prepared the paper. W.J. and W.S. supervised the work.

## Competing interests

The authors declare no competing interests.
