## [Peer Review File · Nature Communications]

Spatiotemporal proteomic atlas of multiple brain regions across early fetal to neonatal stages in cynomolgus monkeyREVIEWER COMMENTS

Reviewer #1 (Remarks to the Author):

The manuscript generated a spatiotemporal proteomic atlas of cynomolgus macaque brains from early fetal to neonatal stages. The study provided a precious resource, however, the current version of manuscript did not demonstrate much new biological insights revealed by current work compared with previous transcriptomic and proteomic studies.

The main findings of the study, including the regional differences and the temporal variations, has been revealed by previous transcriptomic analysis in human and rhesus macaques (ref. 43). Besides, the conclusion that human proteomic profile is closer to primates than to mice seems not surprising given their genetic divergence. Further analysis to discover new biological findings would be appreciated. Analysis could be especially expanded on protein specific aspects. For example, the spatiotemporal patterns of different protein families or proteins with different subcellular locations could be explored. In addition, the difference between gene and protein expression could be carefully investigated to uncover potential post-translational regulations.

In the cross-species comparison part, while it is expected that cynomolgus macaque is closer to human than mice globally, it is still interesting to investigate what proteins, protein families, and pathways are conserved across species; what are human specific; and what are different between primates and mice. Some other comments:

1. The cross-species comparisons maybe affected by unmatched developmental stages between data sets. The authors claimed that “as brain development progressed from F90 to P3, the monkey proteome became more similar to the human”. It may be because the human proteome data used for comparison are from postnatal brains. Besides, the mouse proteomic dataset generated by Kirti et al includes cerebellar tissue at different postnatal time points, and other brain regions from adult.
2. Yellow circles in Figure 2A are not explained in figure legends.
3. The label of Figure 2D x-axis should be “F50”, “F90”and“F120”.
4. The biological interpretation of the observations was not fully discussed. For example, why the reflection point of cerebellum is different from other brain regions? What do regional differences in disease genes imply?

Reviewer #2 (Remarks to the Author):

In this study authors conducted a proteomics analysis of multiple brain regions at four developmental stages, three of them prenatal, in macaques.

The study is creating a valuable data resource, as systematic brain proteome data in primates is rare. The analysis and its interpretation, however, do not match the data’s potential.

One of the first issues is lack of described protein detection criteria. MS-based proteomics yields peptide counts data with most proteins detected by one or two peptide and each peptide frequently appearing

ones or zero times across samples. Such sparse data is not very suitable for common parametric statistical tests. I found no description of the thresholds used. Further, even the description in the study Results section is misleading: line 106 states that 9286 proteins were detected in all 156 samples, giving an impression that they were detected in each of them. Further reading, however, shows that it is not true, as proteins called detected in stages or regions are fewer.

The statistical tests is another issue. The methods simply states that ANOVA with BF correction was used, but it seems unlikely given that more than 2/3 of detected proteins are called significant. Further, results shown in Fig. 1D are unlikely to be coming from the ANOVA, as ANOVA does not provide information about pairwise differences among regions. If 1D is based on t-test, then there must be no multiple testing correction, since 3x3 sample comparison cannot result in so many statistically valid differences between brain regions simply due to power issues.

These two issues alone do not allow to estimate the validity of results presented by the authors, The very minimal initial steps that should be taken by the authors is:

1. clearly state the detection criteria for proteins
2. Depending on these criteria, the authors should choose appropriate statistics: If they use very strict thresholds allowing no missing count values for any proteins in each of the samples, then log-transformed sound could potentially be used in parametric tests, but still would require permutation tests to verify the significance and gauge the FDR. If that go for sparse data with loose detection cutoff, as I believe they did now based on detected protein numbers, completely different statistical tools should be used.

The trouble with an entire manuscript is that authors appear to disregard the very possibility that most of their observations could be random effects of big data analysis and simply describe it in a very direct way with no thought for verification or, often, logic. Below, I will list some examples illustrating these points:

Lines 112-115 Authors simply mention the largest numbers in a pairwise comparison table without testing whether this region/stage excess of differences is significant. Further, as mentioned above, most likely the numbers could not be taken for their face value because of statistical analysis issues (no multiple testing correction and use of parametric tests for count sparse data).

Lines 117-118 (Fig 1E result) the authors claim seeing “different patterns” providing no statistical evidence for the difference. It could be to some extent acceptable if the differences were at least visually evident, but they are definitely not.

Lines 123-125 The correlation analysis simply shows that adjacent stages correlate best, separated by one stage - worse, and separated by two stages - the worst. Somehow authors fail to see this and simply report the largest and the smallest correlation. The problem with it, that the largest correlation is likely to be not significantly better than the other two for adjacent stage comparison.

Lines 131-132 The analysis shown in Fig. 2D is not directly relevant to the statement that changes in protein abundance did not represent a continuous increase. Further, data normalisation procedures would affect this, but they are not described either.

Line 135 Not clear how stage-specific marker proteins were defined statistically. If it was done using Seurat, it is a clarification, not a statistical analysis tool. Given the number of samples per stage (three), the results could very well be spurious.

Lines 136-144 The proteins listed seem to be selected at hoc and have no obvious connection with biological process terms listed in Fig. S3. Further, the statistical criteria for Fig S3 analysis are not listed. Were p-values adjusted for multiple testing?

Lines 148-149 PCA is not a good way to assess the extent of the differences, especially among multiple plots. For instance, cerebellum data does look more distinct at P3 stage than at F50, but PC1 of F50 explains 40.5% of the variance, while 22.1% for P3.

Lines 151-152 In the cerebellum - cerebrum comparison authors only look at proteins with higher expression in cerebellum, thus missing out the other half of the differences, While their rationale to find “cerebellum-specific” proteins is clear, these proteins cannot be called specific by any measure. They simply have somewhat higher expression.

The same problems as for cerebellum are generally present in cortex/subcortical region analysis, except here differences between the two are much less obvious and, more likely, just random. Thus statement (lines 171-172) about “dramatic” changes of enriched biological process call for statistical examination of how stable these enrichment results really are.

More generally, for results stating significant difference of enriched biological processes (also lines 182-183) I have no idea how one can possible conduct such a statistics, and it is not explained in the manuscript.

Line 184 “abundant new proteins” this seems baseless and contradicting the detected protein numbers. Further, the definition of “new proteins” is lacking.

Lines 192-198 The IHC analysis needs to be quantified and compared side-by-side using statistical analysis to the protein abundance data. Now it’s just pattern and image, it is suggestive, but definitely not sufficient.

Line 205 Anything starting with “remarkable” should better be supported by a statistical test to show that it is not just a big data fluke.

Lines 211-217 There is no variation shown on the graphs (Fig. 5D). Without it, it is impossible to estimate the robustness of this observation.

Lines 226-229 The analysis based on total protein intensities (counts) is useless, as it also reflects the detection properties of peptides in MS. therefore, authors get such enormous correlations: 0.98-0.95. this should be excluded. The comparison of expression levels profiles of each protein in five regions is fine, it's a good analysis. Just, please state age of human and mouse samples.

Figure 5: B - labels flipped

These are just some obvious points, there are other problems. My suggestion would be to consult experts on such data analysis and re-analyse the whole thing completely, starting from raw data in a good way.

Reviewer #3 (Remarks to the Author):

The manuscript titled "High-resolution spatiotemporal proteomic atlas of multiple brain regions across early fetal to neonatal stages in cynomolgus monkey", describes a proteomic study of three fetal time points (F50, F90, F120) and neonatal (P3) macaque brain. Regional samples vary from 5 (in the earliest fetal samples) to 18 in the neonatal as well as the F120 samples. Protein abundance is measured using 4D proteomics technology and comparisons are made between regions as well as between developmental stages. I think, with a minor revision this manuscript will be a nice addition to the field.

The manuscript is an easy read and the quality sound. There are a few, for me, confusing sentences. And I miss more comparison to the textbook knowledge of development. Some of the findings are not as novel or striking as described and could use further comparisons or rephrasing.

There are some more general comments related to choices of words and phrases that I think should be reconsidered or clarified.

-“High resolution”. I question the use of high resolution in this situation. The analysis is not on cell level, nor sub regional level, it is a tissue sample 0.5-2mm³. And if “high-resolution” is used it should be clarified in what regard or compared to what?

-“Protein expression” is used at several occasions. This is misleading. Detecting proteins with MS is not the same as the expression of the protein. Depending on the protein halftime, the actual expression could potentially happen at a different time and even different location. mRNA and protein does not always correlate, that ratio is gene-specific and the location depends on the secretory pathway. And perhaps depending on how well the projections have developed this probably also varies across developmental stages.

-“Specifically expressed” double issue, expression is not measured here, but protein detection. Secondly, does “specific” refer to under cut off in all samples but that one sample? Is the cut off defined and how was it decided?

Related to the main findings:

-“There is a stage-specific pattern of protein detected”. This should not be a surprise. And could be further discussed. Different stages in neurodevelopment is related to different expression profiles. I would like to see more comparison to literature and textbook information regarding this. That there are differences is expected, so follow up question would be- does the result match expected neurodevelopmental markers in the timeline? There is limited translation into the human weeks and knowledge in the manuscript. Yes, this is the first proteomic level analysis, but there should be literature related to the stage-specific proteins/genes to look at. Do you see the expected textbook markers in the different stages? This will be limited to human orthologues, but in the end that is the whole purpose, right? To translate the findings into human. Lines 181 and 186 made me happy, finally some translation into human week comparisons, but this could be further extended.

-That Cerebellum is different from the other regions is not a surprise, there are numerous publications highlighting that Cerebellum is the brain region that stands out with a unique protein profile. Additionally, in this case when Cerebellum is the only region representing hindbrain it is even less remarkable. To keep that focus I suggest comparing if the cerebellum specific proteins are the same as expected, based on known hindbrain transcription factors, transcriptomic data etc.. The cell types in cerebellum is unique compared the other regions, are the proteins detected related to a certain cell type? Or is more related to the fact that most cerebellar cells are inhibitory? Since “cerebellum standing out” is not a new story I suggest adding an extra layer of follow up on those proteins, cellular location or why not related to genes with connection to medulloblastoma? That surprised me, there is a discussion related using this data to learn about disease, and cerebellum is highlighted but medulloblastoma genes are not explored?

-“Human and monkey are more similar than human and mouse”, is a story I find poorly formulated. This is not a surprise, once again this has been shown by many. How can a proteomics study improve the knowledge? What is the news related to this study and this specific question? Is protein correlation higher or lower than transcriptomic comparison? The sentence (line 241-242) that says monkey brain become more similar to human with the progression of development is very misleading, since the whole comparison is to postnatal human brain. It is like saying the monkey brain is more similar to the postnatal sample the closer to birth you sample.

Other detailed comments:

-It is on several occasions highlight how this is a great resource, and I fully agree, the number of samples and proteins detected are great and will complement the lack of human data. However, it is very unclear how to use and access this data. The links provided are not very informative and seems to be more like supporting table type where both knowledge of the files and bioinformatics is required. If you are serious in making it a useful resource, consider making a portal where protein names are searchable, regions clickable or some other way to display the data in an easier way for users that are not bioinformaticians or lack computer power.

-Line 48 in the introduction, I lack examples of resources, for example Allen Brain Institute provide several related useful tools on this.

-Sentence in line 58-59. I don't understand the purpose with this sentence, have not every animal undergone natural selection. The topic is anyway discussed in line 61-65 so this sentence is just strange.

- Line 109, related to the largest and smallest number of detected proteins in F50 and P3 respectively. How does this suggest decrease in pluripotency? I agree that it is the case, from textbooks. But the protein detected is not normalized to cell types or number of cells, right? Could it not also be that the same size of sample in a P3 brain sample include relatively more of the same cells since it has expanded and is more differentiated. Compared to the F50 that probably have smaller pools of more similar cells. How do you correct for this possibility?
- Three biological samples are mentioned but never shown. I'm curious of the inter-sample variation but can't find any information related to this. Especially important for all the differentially expression comparisons- what is the variation within the same samples first of all?
- In the "393344_0_additional_review_material_0_rj8qzs.pdf" file it says 168 samples... but in the manuscript only 156 samples are mentioned, did I miss the place where this is clarified?
- Table S1 is mentioned once, and not until I opened it did I get information about the sex of the samples (I think sex is the correct term, not gender, in this case). I think it, at least, should be mentioned in the M&M and the fact that all F50 are male and all the F90 are females, did you look at any of the sex-specific genes? Can any of the differentially detected proteins be explained by this factor?
- In figure 3 legend it says three technical replicates were done, I thought it was biological replicates, once again a little confused. And why only from two of the F90 CB samples?
- I miss some information in the material and method. Where are the details and criteria of several of the analysis described? Cut off for a specific expression, biological processes-information extraction. Did I miss where the definition of DEP is described?
- Line 284 "Proteins are the structural basis of cells and execute neural activities" I do not understand the purpose of this sentence.
- Where are the H&E stainings? I find no comments or images of those.

Comments related to figure and figure legends:

- In several cases I miss details in the figure, explaining the color code. Or in the legend explaining how the values were calculated.
- The first reference to Fig1A is confusing to me (line 92), that sentence is not really related to the figure
- Figure 1 title is not that representative. Where is the resource overview? I see a sampling overview, and detection number overviews. A good figure 1 but the title does not really match for me.
- Figure 2- it never states what the yellow ring in A is, I guess those are the CB samples, based on text later in the manuscript, but it took a while to get to that information.
- Figure 2, I don't understand what B shows, is it only showing the CB specific proteins, not that specific in that case, and the total proteins? i.e. not comparable? What is the purpose, just to show that there are a group of proteins with higher expression in CB?
- Figure 4, perhaps the background of NCAN and SOX2- could be mentioned, and referred to the supporting image explaining the selection.
- Figure 5, line 201 mentioned fig 5A, the PCA plots as proteomic variation. Isn't PCA more of clustering analysis, a visualization tool? To talk about variation shouldn't a computational analysis be done?

REVIEWER COMMENTS

Reviewer #1 (Remarks to the Author):

The manuscript generated a spatiotemporal proteomic atlas of cynomolgus macaque brains from early fetal to neonatal stages. The study provided a precious resource, however, the current version of manuscript did not demonstrate much new biological insights revealed by current work compared with previous transcriptomic and proteomic studies.

The main findings of the study, including the regional differences and the temporal variations, has been revealed by previous transcriptomic analysis in human and rhesus macaques (ref. 43). Besides, the conclusion that human proteomic profile is closer to primates than to mice seems not surprising given their genetic divergence.

Further analysis to discover new biological findings would be appreciated. Analysis could be especially expanded on protein specific aspects. For example, the spatiotemporal patterns of different protein families or proteins with different subcellular locations could be explored. In addition, the difference between gene and protein expression could be carefully investigated to uncover potential post-translational regulations.

Response: We appreciate the reviewer's insightful suggestions for further analysis to gain more biological significances from this resource.

As suggested, we analyzed and depicted the spatiotemporal changes of different protein families from the fetal to postnatal stages. Along with fetal brain development, we found the changes of the protein families were different and showed three major patterns. The detailed results were shown in Figure 1F and were described in Line 123-130 in the revised manuscript.

We have further analyzed subcellular localization of the proteins in different brain regions and added statistical significance analysis of the changes in subcellular localizations as fetal brain development. We found a high degree of similarity among different regions of the cortex. We characterized the regulated change trend of Nucleoplasm, plasma membrane, and cytosol proteins. The detailed results were shown in Figure 1E and were described in Line 117-122 in the revised manuscript.

As the reviewer's suggestion, we compared the changes of our proteomic data from prenatal (F120) and postnatal (P3) monkeys to the previously published monkey transcriptomic data (ref. 12) from prenatal (110 days after fertilization) and postnatal (2 days after birth) stages. Based on the FC of RNA and protein levels, all genes were divided into 6 types (Fig. 6D). Type 1-3 genes accounted for more than 90% of the all genes. The results highlighted a certain proportion (>15%) of genes with mRNA/protein discrepancies. We further elucidated the enrichment pathways of different gene types. The result emphasizes the importance of proteomic studies of brain development and is an important complement to previous transcriptomic studies. The details were shown in Figure 6D, Figure S6-8 and were described in Line 285-303 in the revised manuscript.

In the cross-species comparison part, while it is expected that cynomolgus macaque is

closer to human than mice globally, it is still interesting to investigate what proteins, protein families, and pathways are conserved across species; what are human specific; and what are different between primates and mice.

Response: Many thanks for this inspiring suggestion. We performed further analysis for the cross-species comparison. Based on correlation analysis, three classes of proteins were identified, which were defined as conserved proteins, human-specific proteins and primate-specific proteins. By enrichment analysis, we found that the conserved proteins were mainly enriched in dendritic spine development, RNA splicing, etc., the human-specific proteins were mainly enriched in Golgi vesicle transport, macroautophagy, etc. and the primate-specific proteins were mainly enriched in neuron development, anterograde trans-synaptic signaling, etc. The classification of these proteins helps us to understand species differences in brain development and evolution at the protein level. The detailed results were shown in Figure 6C and were described in Line 275-284 in the revised manuscript.

Some other comments:

1. The cross-species comparisons maybe affected by unmatched developmental stages between data sets. The authors claimed that “as brain development progressed from F90 to P3, the monkey proteome became more similar to the human”. It may be because the human proteome data used for comparison are from postnatal brains. Besides, the mouse proteomic dataset generated by Kirti et al includes cerebellar tissue at different postnatal time points, and other brain regions from adult.

Response: Thanks. As the reviewer pointing out, the matching of tissue sampling time points is very important and determines the reliability of the conclusions in cross-species comparisons. We removed the results of Figure 6C in the original MS due to the inappropriate comparison.

In the revised manuscript, considering no available datasets of mouse and human exactly matching the developmental time points of our monkey dataset, we only used the proteomic data of 3 days postnatal of monkey to perform comparison with the available proteomic data at similar developmental stages of human (ref. 18 by Carlyle et al.) and mouse (ref. 26 by Kirti et al.). The three time points cross three species are all corresponding to postnatal stage. The detailed results were shown in revised Figure 6A-C and were described in Line 256-284 and 579-596 in the revised manuscript.

In addition, we compared the changes of our proteomic data from prenatal (F120) and postnatal (P3) monkeys to the previously published monkey transcriptomic data (ref. 12) from prenatal (110 days after fertilization) and postnatal (2 days after birth) stages. Based on the FC of RNA and protein levels, all genes were divided into 6 types (Fig. 6D). Type 1-3 genes accounted for more than 90% of the all genes. The results highlighted a certain proportion (>15%) of genes with mRNA/protein discrepancies. We further elucidated the enrichment pathways of different gene types. The result emphasizes the importance of proteomic studies of brain development and is an

important complement to previous transcriptomic studies. The details were shown in Figure 6D, Figure S6-8 and were described in Line 285-303 in the revised manuscript.

2. Yellow circles in Figure 2A are not explained in Figure legends.

Response: Thanks. The yellow circle indicated the samples of cerebellum. We added the description in the Figure 2A legend of the revised manuscript.

3. The label of Figure 2D x-axis should be “F50”, “F90”and“F120”.

Response: Thanks. We corrected this typing error in Figure 2D in the revised manuscript.

4. The biological interpretation of the observations was not fully discussed. For example, why the reflection point of cerebellum is different from other brain regions? What do regional differences in disease genes imply?

Response: Thanks. We added some new analysis of the cerebellum in the revised manuscript. The Euclidean Distance analysis quantified the differences between cerebellum and other brain regions (Figure 3B). We identified the upregulated proteins as cerebellum marker proteins at each stage (Figure 3C). Among these up-regulated proteins, GRID2IP and PCP2 are known as the marker proteins of Purkinje cells, and ZIC1 and GABRA6 are markers as granule cells. The abundance of GRID2IP, PCP2, and ZIC1 increased from F90, and GABRA6 increased lately from F120, which might indicate the differentiation of the two cell types in CB (Figure 3D,E). Also, we analyzed the changes of proteins (APC, CTNNB1, DDX3X, PTEN, BCOR) linked to medulloblastomas in cerebellum at four developmental stages (Figure S4). No significant change between stages was found. These results together with the findings of an earlier reflection point in CB (Figure 2D) suggest that neural differentiation in the cerebellum might start at F90, which is earlier than neural differentiation in the cerebrum. The detailed description of the results in Line 163-185 in the revised manuscript.

By referring previous relevant studies, we disused these findings as follows.

“In the comparison of CB and other brain regions, the proteomic data revealed that CB had a distinct developmental pattern and showed increasing Euclidean distances along with brain development. The distance between the CB and V1 did not increase with development. We identified the up-regulated proteins as cerebellum marker proteins at each stage. Enrichment analysis of the marker proteins emphasizes distinct biological processes and pathways at different developmental stages. For example, the enrichment of mRNA processing related processes was observed both at F120 and P3 stages, which suggests that trends in the CB at late fetal stages continue after birth. The result was consistent with information from postnatal human brain¹⁸, suggesting a specialization of CB during the perinatal period in primates. The GRID2IP, PCP2, ZIC1 and GABRA6

were observed to highly express at F120 and P3 stages. These proteins were known to related to the maturation of purkinje cells and granule cells^{28,29}, respectively. The above results together with the findings that CB had an earlier reflection point of inter-regional differences suggest that neural differentiation in the cerebellum might start at F90, which is earlier than neural differentiation in the cerebrum. (Line 324-338)

We have added statistical analysis of protein changes in disease risk genes. Proteins of risk genes linked to neuropsychiatric diseases were significantly changed in the PFC during the fetal brain development. In contrast, the changes were not significant in V1. This finding suggests that the functional execution of the PFC may be more sensitive to neuropsychiatric diseases than V1. Up to date, previous studies barely reported the relationship between primary visual cortex (V1) and neurodegenerative diseases. Our data revealed the proteins of risk genes associated with AD and PD changed significantly in V1 during fetal brain development, which implies further investigation on the roles of V1 in the pathogenesis of degenerative diseases deserves more attention. The detailed results were shown in Figure 5D and were described in Line 240-253 in the revised manuscript.

By referring previous relevant studies, we disused these findings as follows.

“Our study, at the protein level, revealed similar synchronous trends in cortical regions, as well as in subcortical regions. It has been reported that the mid- and late fetal stages are a key developmental period in humans and that these stages are also associated with the etiology of neuropsychiatric and neurodegenerative disease^{12,45,46}. Our proteomic data demonstrates that protein expression related to neuropsychiatric and neurodegenerative diseases showed opposite trajectories both in PFC and V1. Proteins of risk genes linked to neuropsychiatric diseases were significantly changed in the PFC during the fetal brain development. In contrast, the changes were not significant in V1. This finding suggests that the functional execution of the PFC may be more sensitive to neuropsychiatric diseases than V1. Up to date, previous studies barely reported the relationship between primary visual cortex (V1) and neurodegenerative diseases. Our data revealed the proteins of risk genes associated with AD and PD changed significantly in V1 during fetal brain development, which implies further investigation on the roles of V1 in the pathogenesis of degenerative diseases deserves more attention. Confirmation of this hypothesis in future work may allow early detection and treatment of these conditions.” (Line 354-370)

Reviewer #2 (Remarks to the Author):

In this study authors conducted a proteomics analysis of multiple brain regions at four developmental stages, three of them prenatal, in macaques.

The study is creating a valuable data resource, as systematic brain proteome data in primates is rare. The analysis and its interpretation, however, do not match the data's potential.

One of the first issues is lack of described protein detection criteria. MS-based proteomics yields peptide counts data with most proteins detected by one or two peptide and each peptide frequently appearing ones or zero times across samples. Such sparse data is not very suitable for common parametric statistical tests. I found no description of the thresholds used. Further, even the description in the study Results section is misleading: Line 106 states that 9286 proteins were detected in all 156 samples, giving an impression that they were detected in each of them. Further reading, however, shows that it is not true, as proteins called detected in stages or regions are fewer.

Response: Thanks for the reviewer's professional comments.

We have added a new section "Protein identification and quantification" in "Material and Methods (M&M)" of the revised manuscript to illustrate the methodologies for protein detection. The description is as follows.

"Protein identification and quantification"

In this study, we used the method of label-free quantification (LFQ) to determine the relative abundance of proteins in our samples. Peptide was identified with the posterior error probability (PEP) score <0.05 and minimum score for modified peptides >40 . False discovery rate (FDR) thresholds for protein and peptide were specified at 1%. Minimum peptide length was set at 7. All other parameters in MaxQuant were set to default values. A total of 33314103 secondary spectra were obtained from the mass spectrometry analysis. After searching the protein theoretical data, the number of usable secondary spectra was 6760839, which is 20.3% of the spectrum utilization rate. A total of 120,730 peptides were identified by spectral analysis, of which 97,173 were unique peptides that match a single protein. The other peptides match multiple proteins, which are called "razor" peptides. The "razor" peptides are assigned to the protein group with the most unique peptides. Then the relative intensity of these peptides was quantified as the area of the peak that was extracted from the primary mass spectrometry (MS1). Finally, MaxQuant employs the MaxLFQ algorithm for label-free quantitation (LFQ)⁵⁰. The intensities of individual proteins will be corrected for quantitative information based on the peptides identified across samples, and the LFQ intensity will be calculated to minimize inter-sample errors caused by handling, loading, pre-sorting, instrumentation, etc., to allow quantitative comparisons of the same proteins across samples. The LFQ intensity >0 means the protein is detected in this sample. Raw results for the identification and quantification of all peptides and proteins are shown in the Table S11. LFQ intensities of proteins were used for downstream analysis and statistics." (Lines 458-480)

We re-edit the description for the results of protein detection to make it more clear for audiences. The description is as follows (Line 107-114).

“The protein abundances were expressed as label-free quantification (LFQ) intensity and log₂ transformed before differential expression analysis. (Fig. 1B and Table S2). In the 156 samples, an average of 5967 proteins were identified per sample (5423-6600 proteins). The union set of identified proteins from the 156 samples was 9286, of which 7618 proteins were annotated with gene information. There was an average of 6002 proteins per stage (5885-6191 proteins) (Fig. 1C and Table S3). The largest and least average number of proteins were identified at F50 and P3, respectively (Fig. 1C).”

The statistical tests is another issue. The methods simply states that ANOVA with BF correction was used, but it seems unlikely given that more than 2/3 of detected proteins are called significant. Further, results shown in Fig. 1D are unlikely to be coming from the ANOVA, as ANOVA does not provide information about pairwise differences among regions. If 1D is based on t-test, then there must be no multiple testing correction, since 3x3 sample comparison cannot result in so many statistically valid differences between brain regions simply due to power issues.

These two issues alone do not allow to estimate the validity of results presented by the authors, The very minimal initial steps that should be taken by the authors is:

1. clearly state the detection criteria for proteins

Response: Thanks. In the new section “Protein identification and quantification” in “M&M” of the revised manuscript, we described the methodologies how to detect and quantify protein from peptides (Lines 458-480). Please find it also in the response above.

2. Depending on these criteria, the authors should choose appropriate statistics: If they use very strict thresholds allowing no missing count values for any proteins in each of the samples, then log-transformed sound could potentially be used in parametric tests, but still would require permutation tests to verify the significance and gauge the FDR. If that go for sparse data with loose detection cutoff, as I believe they did now based on detected protein numbers, completely different statistical tools should be used.

Response: Thanks for the professional advice.

In the new section “Protein identification and quantification” in “M&M”, we noted that it is the LFQ intensity, not the count value, were used for downstream analysis and statistics.

As the reviewer suggested, the student’s t-test with Bonferroni correction was used to calculate the differentially expressed proteins (DEPs) based on the log₂-transformed protein abundance (LFQ intensity). The calculation of LFQ intensity was described in the new section named “Protein identification and quantification” in the revised manuscript (Lines 458-480).

The trouble with an entire manuscript is that authors appear to disregard the very possibility

that most of their observations could be random effects of big data analysis and simply describe it in a very direct way with no thought for verification or, often, logic. Below, I will list some examples illustrating these points:

Response: In this study, the samples were collected from three biological replicates at each developmental stage, which allowed us to make statistical comparisons between any individual brain region at any time point. From the reviewer's comment, the authors have realized that some descriptions of the comparative results in the original manuscript missed to provide statistical information, such as mean, SD, p-value. In the revised manuscript, the differences of protein abundance (LFQ intensity) among groups were calculated based on a rigorous statistical approach (student's t-test with Bonferroni correction). We added statistical information in the new Figures including Figures 1b-f, 2b-d, 3c-e, 4e, 5b-d. Also, we have made response point by point to reviewer's specific concerns below.

Lines 112-115 Authors simply mention the largest numbers in a pairwise comparison table without testing whether this region/stage excess of differences is significant. Further, as mentioned above, most likely the numbers could not be taken for their face value because of statistical analysis issues (no multiple testing correction and use of parametric tests for count sparse data).

Response: Thanks. For the Fig 1D, the student's t-test with Bonferroni correction was used to calculate the differentially expressed proteins (DEPs) based on the log₂-transformed protein abundance (LFQ intensity). We added the description of calculation of DEPs (Lines 502-506), and modified the description of the result in the revised manuscript (Lines 114-116).

“In same stage, differentially expressed proteins (DEP) in a specific region are identified by comparing the differences between the specific region and all other regions using t-test with Bonferroni correction. The protein with adjusted p-value <0.05 was consider as DEP (**Figure 1D**).” (Lines 502-506)

“The differentially expressed proteins (DEPs) was calculated by the student's t-test with Bonferroni correction. The number of DEPs of different regions at the four stages were shown in Fig. 1D.” (Lines 114-116)

Lines 117-118 (Fig 1E result) the authors claim seeing “different patterns” providing no statistical evidence for the difference. It could be to some extent acceptable if the differences were at least visually evident, but they are definitely not.

Response: Thanks. We re-analyzed subcellular localization of the proteins in different brain regions and added statistical significance analysis of the changes in organelle located proteins as fetal brain development. The significance of differences between adjacent stages was calculated by using the student's t-test. The detailed results were shown in Figure 1E, Table S4 and were described in Line 117-122 in the revised

manuscript.

Lines 123-125 The correlation analysis simply shows that adjacent stages correlate best, separated by one stage - worse, and separated by two stages - the worst. Somehow authors fail to see this and simply report the largest and the smallest correlation. The problem with it, that the largest correlation is likely to be not significantly better than the other two for adjacent stage comparison.

Response: Thanks. As suggested, the Pearson correlation between stages were recalculated by using `cor` and `cor.test` in R language in the revised manuscript. The correlation coefficient and significance were shown in Figure 2B. The description of the result was modified as follows.

“The correlation analysis revealed that all correlation between any stages were significant ($P < 0.001$), and the highest correlation (0.89) of adjacent stages was observed between F90 and F120 (Fig. 2B).” (Lines 135-137)

Lines 131-132 The analysis shown in Fig. 2D is not directly relevant to the statement that changes in protein abundance did not represent a continuous increase. Further, data normalisation procedures would affect this, but they are not described either.

Response: Thank for the comments. We deleted the inappropriate description and rewrote the statement (Lines 145-147). The calculation was described as follows in the revised manuscript (Lines 531-536).

“Interestingly, further trend analysis revealed that the inter-regional changes of protein abundance exhibited an overall "V" shaped pattern which bottomed at the F90 in CB or at the F120 stage in CTX and sCTX (Fig. 2D).” (Lines 145-147).

“Next, we grouped samples by regions (CB, cortical areas, subcortical areas) and calculated the inter-regional differences at each developmental stage (Figure 2D). The inter-regional differences were calculated as the average of absolute difference of \log_2 transformed LFQ intensities of paired proteins from all pairs of samples in same grouped area. The significance of differences between adjacent time points was calculated using t-test.” (Lines 521-526)

Line 135 Not clear how stage-specific marker proteins were defined statistically. If it was done using Seurat, it is a clarification, not a statistical analysis tool. Given the number of samples per stage (three), the results could very well be spurious.

Response: Thanks. The stage marker proteins in each stage were identified using the function “`FindAllMarkers`” with default parameters except for `test.use = "roc"`. Proteins with $AUC > 0.7$ and fold change > 1 was considered marker proteins. The top 10 marker proteins sorted by AUC value for each stage were shown as heatmap (Fig 2E). In order to find the stage marker proteins, we grouped the samples of all regions (CB excluded) at each stage, and the number of samples at the four stages for statistical comparison

were 12 (F50), 30 (F90), 51 (F90) and 51(P3), respectively. The calculation was described in the revised manuscript (Lines 526-532).

Lines 136-144 The proteins listed seem to be selected at hoc and have no obvious connection with biological process terms listed in Fig. S3. Further, the statistical criteria for Fig S3 analysis are not listed. Were p-values adjusted for multiple testing?

Response: The stage marker proteins shown in Figure 2E were the top 10 ranked by AUC and FC. In new Fig. S2 (corresponding to the original Figure S3), the list was the enrichment result of all marker at each stage, and only the top 5 enriched biological processes were showed for each stage. MCM2-7 proteins were up-regulated at the F50 stage. These proteins are play roles in the biological process of DNA replication (The adjusted P value is $2.92E-14$ and the rank is 17 based on enrichment result). The DNA replication was also enriched in the KEGG pathway (new Fig. S2B). We showed the p-values adjusted for multiple testing for the enriched biological processes and pathways (new Fig. S2).

Lines 148-149 PCA is not a good way to assess the extent of the differences, especially among multiple plots. For instance, cerebellum data does look more distinct at P3 stage than at F50, but PC1 of F50 explains 40.5% of the variance, while 22.1% for P3.

Response: Thanks. Yes, the PCA is more of clustering analysis, not a good way to represent distances. In the revised manuscript, the distance between CB and other brain regions was measured in Euclidean distance and calculated by using dist function in R language. The detailed results were shown in Figure 3B and the description of this result was adjusted as follows.

“The results showed that protein profiling in the CB was distinct from the other brain regions at each stage (Fig. 3A), and the dissimilarity increased over development except V1 region (Fig. 3B).” (Lines 162-165).

Lines 151-152 In the cerebellum - cerebrum comparison authors only look at proteins with higher expression in cerebellum, thus missing out the other half of the differences, While their rationale to find “cerebellum-specific” proteins is clear, these proteins cannot be called specific by any measure. They simply have somewhat higher expression.

Response: Thanks. We have revised the original Figure 3B. The term of “cerebellum-specific” proteins was changed as “CB marker proteins” to represent up-regulated proteins in cerebellum significantly. The CB marker proteins were identified using the function “FindAllMarkers” default parameters except for test.use = "roc". Protein with AUC >0.7 and fold change >1 was considered CB marker proteins. After identification of CB marker proteins, the top 20 marker proteins sorted by AUC value for each stage were shown as heatmap (Figure 3B).

The same problems as for cerebellum are generally present in cortex/subcortical region

analysis, except here differences between the two are much less obvious and, more likely, just random. Thus statement (Lines 171-172) about “dramatic” changes of enriched biological process call for statistical examination of how stable these enrichment results really are.

Response: Thanks. We have added detailed description as follows in the revised manuscript.

“Here we focus on describing the calculation of the overlap rate of biological processes in Figure 4C, which refers to the previous study². For cortical and subcortical, the marker protein was first calculated separately for the two regions at each stage. Then the biological processes were enriched based on the marker protein at each stage. When the biological processes for each stage were obtained, the similarity of the biological processes between two adjacent developmental stages was calculated. The newly enriched biological processes at the indicated stages were identified by comparison of the biological processes at the pre-stage of this stages. The newly enriched biological processes in cortical and subcortical regions were calculated and shown in the Figure 4D, respectively.” (Lines 543-552)

We have changed the statement about “dramatic” changes as follows.

“The analysis of the overlap of biological processes enriched by marker proteins between two adjacent stages showed the dynamic changes during fetal brain development (Fig. 4C and Table S9).” (Lines 197-199).

More generally, for results stating significant difference of enriched biological processes (also Lines 182-183) I have no idea how one can possible conduct such a statistics, and it is not explained in the manuscript.

Response: we apologize for the inappropriate description. It is impossible to conduct this statistic. We re-wrote the description as follows.

“Unlike other stages, the cortical and subcortical regions showed unmatched biological processes at F120.” (Lines 208-210).

Line 184 “abundant new proteins” this seems baseless and contradicting the detected protein numbers. Further, the definition of “new proteins” is lacking.

Response: Thank you for your comments. We apologize for the unclear description in the results. We have changed it as follows.

“The results suggest that new biological processes might emerge to adapt to changes in brain structures and functions at F120 stage, which corresponds to 32 weeks post-fertilization in humans.” (Lines 210-212).

Lines 192-198 The **IHC analysis** needs to be quantified and compared side-by-side using statistical analysis to the protein abundance data. Now it’s just pattern and image, it is suggestive, but definitely not sufficient.

Response: Thanks for suggestion. However, due to the difficulty in obtaining fetus and the ethical limits, the samples obtained were used for proteomic with priority. Furthermore, the tissues collected from F50 and F90 were very small. Therefore, the brain tissues for histological staining could not support enough statistical repeats. The brain IHC only showed the distribution of these proteins.

Line 205 Anything starting with “remarkable” should better be supported by a statistical test to show that it is not just a big data fluke.

Response: Thanks. We have added statistical significance tests as well as Means and SD (Figure 5B, C). We have revised the relevant content based on the new results (Lines 231-240).

Lines 211-217 There is no variation shown on the graphs (Fig. 5D). Without it, it is impossible to estimate the robustness of this observation.

Response: Thanks. We have added statistical significance tests as well as means and variances in new Figure 5D. The relevant description of the result was modified basing on the new Figure (Lines 245-251).

Lines 226-229 The analysis based on total protein intensities (counts) is useless, as it also reflects the detection properties of peptides in MS. therefore, authors get such enormous correlations: 0.98-0.95. this should be excluded. The comparison of expression levels profiles of each protein in five regions is fine, it's a good analysis. Just, please state age of human and mouse samples.

Response: Thanks. About protein detection, we have added a new section called “Protein identification and quantification” in the revised manuscript to describe in detail how to detect and quantify protein from peptides (Lines 458-480).

Considering no available datasets of mouse and human exactly matching the developmental time points of our monkey dataset, we only used the proteomic data of 3 days postnatal of monkey to perform comparison with the proteomic data at similar developmental stages of human (ref. 18 by Carlyle et al.) and mouse (ref. 26 by Kirti et al.). The three time points cross three species are all corresponding to postnatal stage. The detailed results were shown in revised Figure 6C and were described in Line 275-284 in the revised manuscript.

Figure 5: **B - labels** flipped

Response: Thanks. We have redrawn the Figure 5 and corrected the error.

These are just some obvious points, there are other problems. My suggestion would be to consult experts on such data analysis and re-analyse the whole thing completely, starting

from raw data in a good way.

Response: Thanks. We have reanalyzed and explained each of the points raised by the reviewer. The methods for protein identification and quantification and statistical analysis mentioned by the reviewer are described in detail in the Methods section. (Lines 458-615). We added statistical information in the new Figures including Figures 1b-f, 2b-d, 3c-e, 4e, 5b-d.

Reviewer #3 (Remarks to the Author):

The manuscript titled “High-resolution spatiotemporal proteomic atlas of multiple brain regions across early fetal to neonatal stages in cynomolgus monkey”, describes a proteomic study of three fetal time points (F50, F90, F120) and neonatal (P3) macaque brain. Regional samples vary from 5 (in the earliest fetal samples) to 18 in the neonatal as well as the F120 samples. Protein abundance is measured using 4D proteomics technology and comparisons are made between regions as well as between developmental stages. I think, with a minor revision this manuscript will be a nice addition to the field.

The manuscript is an easy read and the quality sound. There are a few, for me, confusing sentences. And I miss more comparison to the textbook knowledge of development. Some of the findings are not as novel or striking as described and could use further comparisons or rephrasing.

Response: Thanks. We modified the manuscript and improved the readability and significance. More important, based on the comments and suggestions of the reviewer, we further analyzed cerebellum characteristic and disease related proteins. In addition, we performed cross-species proteomic comparison and analyzed the difference between proteomic and transcriptomic data of monkey brain. We discussed the observations, which might be beneficial for the deep understanding of early brain development.

There are some more general comments related to choices of words and phrases that I think should be reconsidered or clarified.

-“High resolution”. I question the use of high resolution in this situation. The analysis is not on cell level, nor sub regional level, it is a tissue sample 0.5-2mm³. And if “high-resolution” is used it should be clarified in what regard or compared to what?

Response: The term of “high-resolution” means a state-of-the-art 4D-Proteomics technology that can detect more proteins than usual techniques. To avoid confusion, we changed the title as “Spatiotemporal proteomic atlas of multiple brain regions across early fetal to neonatal stages in cynomolgus monkey”.

-“Protein expression” is used at several occasions. This is misleading. Detecting proteins with MS is not the same as the expression of the protein. Depending on the protein halftime, the actual expression could potentially happen at a different time and even different location. mRNA and protein does not always correlate, that ratio is gene-specific and the location depends on the secretory pathway. And perhaps depending on how well the projections have developed this probably also varies across developmental stages.

Response: We agree with the opinion. The “Protein expression” was substituted with “protein abundance” in the revised manuscript. The protein abundance is quantified as LFQ intensity. We have added a new section “Protein identification and quantification” in “Material and Methods (M&M)” of the revised manuscript to illustrate the methodologies for protein detection. The description is as follows (Lines 458-480).

-“Specifically expressed” double issue, expression is not measured here, but protein detection. Secondly, does “specific” refer to under cut off in all samples but that one sample? Is the cut off defined and how was it decided?

Response: Thanks for the suggestion. The “Specifically expressed” was substituted with “marker proteins”, which referred to the up-regulated proteins of specific stage or region. Marker proteins (up-regulated proteins) is quantified as LFQ intensity and defined with p value <0.05 and $fc >1$. The detailed description of the method was shown in Line 468-503 in the revised manuscript.

Related to the main findings:

-“There is a stage-specific pattern of protein detected”. This should not be a surprise. And could be further discussed. Different stages in neurodevelopment is related to different expression profiles. I would like to see more comparison to literature and textbook information regarding this. That there are differences is expected, so follow up question would be- does the result match expected neurodevelopmental markers in the timeLine? There is limited translation into the human weeks and knowledge in the manuscript. Yes, this is the first proteomic level analysis, but there should be literature related to the stage-specific proteins/genes to look at. Do you see the expected textbook markers in the different stages? This will be limited to human orthologues, but in the end that is the whole purpose, right? To translate the findings into human. Lines 181 and 186 made me happy, finally some translation into human week comparisons, but this could be further extended.

Response: In our study, based on the spatiotemporal proteomic atlas, we characterized the proteomic dynamics of cerebellum, cortical and subcortical regions of cerebrum.

In the revised manuscript, several important questions in neurodevelopment are well quantified, such as the spatiotemporal differences in proteomic, the regional differences, the specificity of the cerebellum, the cross-species differences and the difference between proteomic and transcriptomic data of monkey brain.

Carlyle et al. found the variabilities among brain regions were greater than those across infancy to adult stages based on human proteomic data. In contrast, we found a opposite situation in fetus stages. The transcriptomic study on human by Kirti et al. revealed that the reflection time point of inter-regional differences spans the late fetal period to early infancy. However, our proteomic data suggest that the reflection time point of inter-regional differences occurs at the F90 or F120, which was earlier than the finding from Kirti’s transcriptomic data. The result revealed the mRNA/protein discrepancies, which highlights the usefulness of our study to understand human brain development since the

human fetal brain proteome is unavailable so far.

Besides above results, several novel findings have been emphasized in this study:

1) We analyzed the changes of protein families in different brain areas during fetal brain development. There are three major patterns in cortical areas including continuous ascending (such as Protein kinase superfamily and Spectrin family), continuous descending (such as ALB/AFP/VDB family and Intermediate filament family) and descending followed by ascending (such as Tubulin family and Metallo-dependent hydrolases superfamily) (Fig. 1F and Table S5). These results elucidate the dynamic changes of several protein families during brain development, which may help to identify key factors in early brain development at the protein level. (Line 123-130)

2) We observed that cerebellum showed a distinct developmental pattern and identified the upregulated proteins as cerebellum marker proteins at each stage (Figure 3C). Among these up-regulated proteins, GRID2IP and PCP2 are known as the marker proteins of Purkinje cells, and ZIC1 and GABRA6 are markers as granule cells. The abundance of GRID2IP, PCP2, and ZIC1 increased from F90, and GABRA6 increased lately from F120, which might indicate the differentiation of the two cell types in CB (Figure 3D,E). Also, we analyzed the changes of proteins (APC, CTNNB1, DDX3X, PTEN, BCOR) linked to medulloblastomas in cerebellum at four developmental stages (Figure S4). These results together with the findings of an earlier reflection point in CB (Figure 2D) suggest that neural differentiation in the cerebellum might start at F90, which is earlier than neural differentiation in the cerebrum. The detailed description of the results in Line 163-185 in the revised manuscript.

3) We analyzed the proteins of risk genes linked to neurological diseases. Proteins linked to neuropsychiatric diseases were significantly changed in the PFC during the fetal brain development. In contrast, the changes were not significant in V1. This finding suggests that the functional execution of the PFC may be more sensitive to neuropsychiatric diseases than V1. Up to date, previous studies barely reported the relationship between primary visual cortex (V1) and neurodegenerative diseases. Our data revealed the proteins of risk genes associated with AD and PD changed significantly in V1 during fetal brain development, which implies further investigation on the roles of V1 in the pathogenesis of degenerative diseases deserves more attention. The detailed results were shown in Figure 5D and were described in Line 240-253 in the revised manuscript.

4) We performed cross-species comparisons by using the proteomic data of 3 days postnatal of monkey and the available published proteomic data at postnatal developmental stages of human (ref. 18 by Carlyle et al.) and mouse (ref. 26 by Kirti et al.). The three time points cross three species are all corresponding to postnatal stage. We identified three classes of proteins, including species-conserved proteins, human-specific proteins and primate-specific proteins, and their involved biological processes. This result provides detailed information for in deep understanding of species difference. The detailed results were described in Line 256-284 in the revised manuscript.

5) We compared the changes of our proteomic data from prenatal (F120) and postnatal

(P3) monkeys to the previously published monkey transcriptomic data (ref. 12) from prenatal (110 days after fertilization) and postnatal (2 days after birth) stages. Based on the FC of RNA and protein levels, all genes were divided into 6 types (Fig. 6D). Type 1-3 genes accounted for more than 90% of the all genes. The results highlighted a certain proportion (>15%) of genes with mRNA/protein discrepancies. We further elucidated the enrichment pathways of different gene types. The result emphasizes the importance of proteomic studies of brain development and is an important complement to previous transcriptomic studies. The details were shown in Figure 6D, Figure S6-8 and were described in Line 285-303 in the revised manuscript.

-That Cerebellum is different from the other regions is not a surprise, there are numerous publications highlighting that Cerebellum is the brain region that stands out with a unique protein profile. Additionally, in this case when Cerebellum is the only region representing hindbrain it is even less remarkable. To keep that focus I suggest comparing if the cerebellum specific proteins are the same as expected, based on known hindbrain transcription factors, transcriptomic data etc.. The cell types in cerebellum is unique compared the other regions, are the proteins detected related to a certain cell type? Or is more related to the fact that most cerebellar cells are inhibitory? Since “cerebellum standing out” is not a new story I suggest adding an extra layer of follow up on those proteins, cellular location or why not related to genes with connection to medulloblastoma? That surprised me, there is a discussion related using this data to learn about disease, and cerebellum is highlighted but medulloblastoma genes are not explored?

Response: Thanks. As suggested by the reviewer, we identified the upregulated proteins as cerebellum marker proteins at each stage (Figure 3C). Among these up-regulated proteins, GRID2IP and PCP2 are known as the marker proteins of Purkinje cells, and ZIC1 and GABRA6 are markers as granule cells. The abundance of GRID2IP, PCP2, and ZIC1 increased from F90, and GABRA6 increased lately from F120, which might indicate the differentiation of the two cell types in CB (Figure 3D,E). (Line 175-181).

In addition, we analyzed the changes of proteins (APC, CTNNB1, DDX3X, PTEN, BCOR) linked to medulloblastomas in cerebellum at four developmental stages (Figure S4). No significant change between stages was found. The detailed description of the results Line 181-185 in the revised manuscript.

-“Human and monkey are more similar than human and mouse”, is a story I find poorly formulated. This is not a surprise, once again this has been shown by many. How can a proteomics study improve the knowledge? What is the news related to this study and this specific question? Is protein correlation higher or lower than transcriptomic comparison? The sentence (Line 241-242) that says monkey brain become more similar to human with the progression of development is very misleading, since the whole comparison is to postnatal human brain. It is like saying the monkey brain is more similar to the postnatal sample the closer to birth you sample.

Response: As the reviewer pointing out, the matching of tissue sampling time points is very important and determines the reliability of the conclusions in cross-species comparisons. We removed the results of original Figure 6C due to the inappropriate comparison.

We thank the reviewer for the inspiring comments. In the revision, we performed further analysis for the cross-species comparison. Based on correlation analysis, three classes of proteins were identified, which were defined as conserved proteins, human-specific proteins and primate-specific proteins. By enrichment analysis, we found that the conserved proteins were mainly enriched in dendritic spine development, RNA splicing, etc., the human-specific proteins were mainly enriched in Golgi vesicle transport, macroautophagy, etc. and the primate-specific proteins were mainly enriched in neuron development, anterograde trans-synaptic signaling, etc. The classification of these proteins helps us to understand species differences in brain development and evolution at the protein level. The detailed results were shown in Figure 6C and were described in Line 275-284 in the revised manuscript.

In addition, we compared the changes of our proteomic data from prenatal (F120) and postnatal (P3) monkeys to the previously published monkey transcriptomic data (ref. 12) from prenatal (110 days after fertilization) and postnatal (2 days after birth) stages. Based on the FC of RNA and protein levels, all genes were divided into 6 types (Fig. 6D). Type 1-3 genes accounted for more than 90% of the all genes. The results highlighted a certain proportion (>15%) of genes with mRNA/protein discrepancies. We further elucidated the enrichment pathways of different gene types. The result emphasizes the importance of proteomic studies of brain development and is an important complement to previous transcriptomic studies. The details were shown in Figure 6D, Figure S6-8 and were described in Line 285-303 in the revised manuscript.

Other detailed comments:

-It is on several occasions highlight how this is a great resource, and I fully agree, the number of samples and proteins detected are great and will complement the lack of human data. However, it is very unclear how to use and access this data. The links provided are not very informative and seems to be more like supporting table type where both knowledge of the files and bioinformatics is required. If you are serious in making it a useful resource, consider making a portal where protein names are searchable, regions clickable or some other way to display the data in an easier way for users that are not bioinformaticians or lack computer power.

-Line 48 in the introduction, I lack examples of resources, for example Allen Brain Institute provide several related useful tools on this.

Response: Thanks for the suggestion. To better present our data, we have provided an interactive web page (https://cpbra.cn/menu_details/cyno/cBrainDev) for relevant researchers to explore and query the spatiotemporal protein atlas.(Line 634-635)

-Sentence in Line 58-59. I don't understand the purpose with this sentence, have not every animal undergone natural selection. The topic is anyway discussed in Line 61-65 so this sentence is just strange.

Response: Thanks. We have re-organized the background related to the advantages of studying the monkey brain over other animal model for understanding brain development processes. The relevant description was modified as:

“Non-human primates share many genetic and physiological similarities with humans, particularly the central nervous system. Due to limited access to healthy human brain tissue, the macaque (an Old-World monkey) brain is an ideal model for the study of brain development. The neocortical regions of the brain are involved in motor, perceptual and higher cognitive functions²², which are highly developed in primates. In macaques, the neocortex makes up 72% of the brain²³. This is close to the 80% in humans. Abnormalities in the neocortex are highly correlated with psychiatric and neurodegenerative diseases²⁴.” (Lines 56-63).

-Line 109, related to the largest and smallest number of detected proteins in F50 and P3 respectively. How does this suggest decrease in pluripotency? I agree that it is the case, from textbooks. But the protein detected is not normalized to cell types or number of cells, right? Could it not also be that the same size of sample in a P3 brain sample include relatively more of the same cells since it has expanded and is more differentiated. Compared to the F50 that probably have smaller pools of more similar cells. How do you correct for this possibility?

Response: Thanks. We agree with the reviewer's comments. Bulk omics cannot resolve specific cell types and is difficult to quantify. We removed inappropriate presumptions from the results in the revised manuscript.(Line 112-113)

-Three biological samples are mentioned but never shown. I'm curious of the inter-sample variation but can't find any information related to this. Especially important for all the differentially expression comparisons- what is the variation within the same samples first of all?

Response: Thanks. To show the variation of the three biological samples in each brain region, we re-created the new Figure1B-C. These new Figures showed of the total protein abundance and identified proteins. Also, the new Figure 3D-E in the revised manuscript showed the average and variations of three biological samples in cerebellum.

-In the “393344_0_additional_review_material_0_rj8qzs.pdf”file it says 168 samples... but in the manuscript only 156 samples are mentioned, did I miss the place where this is clarified?

Response: Thanks. The total number of samples collected in this study was 156. We corrected this number in this file in the revised version.

-Table S1 is mentioned once, and not until I opened it did I get information about the sex of the samples (I think sex is the correct term, not gender, in this case). I think it, at least, should be mentioned in the M&M and the fact that all F50 are male and all the F90 are females, did you look at any of the sex-specific genes? Can any of the differentially detected proteins be explained by this factor?

Response: Thanks. Due to the difficulty in obtaining samples, the number of samples collected in this study was too small to perform a statistical analysis of sex differences in brain development. The authors think it is possible that the abundance of proteins may differ between the sexes. It requires further validation in a well-controlled study next.

As the reviewer comment, the sex of the samples is an important information. We added the information in the “Materials and Methods” in the revised manuscript (Line 415-417).

-In Figure 3 legend it says three technical replicates were done, I thought it was biological replicates, once again a little confused. And why only from two of the F90 CB samples?

Response: Sorry for not describing it clearly. Three biological repeats were performed on all brain regions of the four stages except CB at F90 stage. As one CB sample was lost at F90, three technical replicates of two CB samples were performed at the F90 stage instead. We added the information in the Figure legend of Figure 3.

-I miss some information in the material and method. Where are the details and criteria of several of the analysis described? Cut off for a specific expression, biological processes-information extraction. Did I miss where the definition of DEP is described?

Response: Thanks. We have added the description of cut off and biological processes - information extraction in the section of “Downstream analysis and statistics” in Materials and Methods (Line 481-615).

The calculation of DEPs was described as follows.

“In same stage, differentially expressed proteins (DEP) in a specific region are identified by comparing the differences between the specific region and all other regions using t-test with Bonferroni correction. The protein with adjusted p-value <0.05 was consider as DEP (**Figure 1D**).” (Lines 502-506)

-Line 284 “Proteins are the structural basis of cells and execute neural activities” I do not

understand the purpose of this sentence.

Response: Thanks. We deleted this sentence in the revised manuscript.

-Where are the H&E stainings? I find no comments or images of those.

Response: Sorry for our mistake. H&E staining was not performed in this study and we removed this part in the revised manuscript.

Comments related to Figure and Figure legends:

-In several cases I miss details in the Figure, explaining the color code. Or in the legend explaining how the values were calculated.

-The first reference to Fig1A is confusing to me (Line 92), that sentence is not really related to the Figure

Response: Sorry for the Typo, we corrected the sentence as follows.

“These timepoints encompass peak periods of neurogenesis and cortical expansion during brain development².” (Line 92-94)

-Figure 1 title is not that representative. Where is the resource overview? I see a sampling overview, and detection number overviews. A good Figure 1 but the title does not really match for me.

Response: Thanks. The title of Figure 1 was revised as “Sampling overview and global proteomic profiles across fetal brain development in cynomolgus monkey”.

-Figure 2- it never states what the yellow ring in A is, I guess those are the CB samples, based on text later in the manuscript, but it took a while to get to that information.

Response: Yes, the yellow circle indicates CB samples. We added the information in the Figure legend.

-Figure 2, I don't understand what B shows, is it only showing the, not that specific in that case, and the total proteins? i.e. not comparable? What is the purpose, just to show that there are a group of proteins with higher expression in CB?

Response: Thanks. The “CB specific proteins” was substituted with “CB marker proteins”, which referred to the up-regulated proteins in CB at each stage.

We re-organized the Figure 3 (related to results of CB). In the new Figure 3C, heatmaps showed the abundance of upregulated proteins in CB at each stage.

-Figure 4, perhaps the background of NCAN and SOX2- could be mentioned, and referred to the supporting image explaining the selection.

Response: Thanks for the suggestion. We added more backgrounds and references of the two proteins. The statement of the sentence was changed as follows.

“It has been well known that SOX2 is a marker of neural stem cells, which play vital roles in pluripotency maintenance ^{30,31}, and NCAN plays a role in synaptic plasticity, neuronal migration and/or formation of axonal fibers ^{32,33}.” (Line 219-221).

-Figure 5, Line 201 mentioned fig 5A, the PCA plots as proteomic variation. Isn't PCA more of clustering analysis, a visualization tool? To talk about variation shouldn't a computational analysis be done?

Response: Sorry for the inappropriate expression. Our aim is to show the clustering of the samples. The statement of the result was modified as follows.

“By UMAP analysis, we found the distributions of the four cortical regions were mixed together at each stage (Fig. 5A), which indicates that the overall protein profiling of cortical regions was similar at the same stage.” (Lines 228-230)

REVIEWER COMMENTS

Reviewer #1 (Remarks to the Author):

The authors have conducted new analyses that have revealed novel findings. However, there is still a lack of biological interpretation of these results. For instance, it is unclear how changes in various protein families may be linked to the development of different brain regions and the interspecies variations.

Furthermore, the study should also address which pathways are enriched in the categories where RNA and protein expression are conserved (type 4 and 5) when comparing the transcriptome and proteome results. Additionally, the potential mechanisms underlying the discrepancy between RNA and protein expression and their biological relevance to brain development and evolution should be discussed.

Reviewer #2 (Remarks to the Author):

The manuscript was modified substantially and i'm happy with the current version. I would still recommend removing Fig. 6A and the corresponding analysis. In my opinion it is an example of bad evolutionary analysis and Kunming is a place with good reputation for evolutionary work.

Reviewer #3 (Remarks to the Author):

Big improvements have been added to the manuscript, new analysis and several new figures. All comments were addressed, and I think the updates improved the manuscript significantly. I would suggest going over the text once more, especially the new additions, to polish some formulations.

I only have a couple minor comments:

Line 106, the phrase "protein expression" is still used to describe the protein detected by MS. The expression of proteins is usually described by mRNA i.e. transcriptomic. The comment from the authors says that they have substituted the use of protein expression, but I still see expression being used on several occasions. Using expression when talking about DEP makes sense, but make sure it is used consistently and with correct intention. Line 246 talks about quantified mean expression, but I assume we are still talking about protein detected? although the change in expression is what leads to change in protein abundance. in summary, stay consistent with the choice and make sure it is conscious.

line 118, "By analysis of proteins with subcellular localizations," My assumption is that all proteins have a subcellular location, or extracellular. When I continue reading, I understand what you are referring to, but this sentence as an introduction provides no details what this is about. Additionally, why would you not find a high degree of similarity? Were you expecting the sub cellular location to be significantly

different between the regions? some type of buildup or hypothesis could improve this part of the story.

Line 124, "the known protein families". Is that in comparison to the unknown protein families? This sentence could be reformulated.

Line 164-165 mentioning that there is an increased distance over development- that is a nice improvement of the CB story, great. Line 182, there are numerous publications related to the differentiation in development of granule cells, there must be some relevant knowledge related to the protein levels of GABRA6 to compare this result with instead of only speculating.

Reviewer comments

Reviewer #1 (Remarks to the Author):

The authors have conducted new analyses that have revealed novel findings. However, there is still a lack of biological interpretation of these results. For instance, it is unclear how changes in various protein families may be linked to the development of different brain regions and the interspecies variations.

Response: Thank for your comments. As your suggestion, we have added biological interpretation of these results in the section of Discuss, for example, the roles of Tubulin family, ALB/AFP/VDB family, and Protein kinase superfamily in brain development.

“We found that the abundance of the protein families changed differently during fetal neurodevelopment, for example, tubulin family, ALB/AFP/VDB family and protein kinase superfamily. The proteins of tubulin family are subunits of microtubules, and different tubulins isoforms may be required for specific microtubule functions, which are crucial for cortical development, including neuronal proliferation, migration and cortical laminar organization. Mutations of tubulin proteins have been identified in individuals with a range of tubulin-related diseases, such as lissencephaly, schizencephaly and microlissencephaly⁴⁵. In this study, we found that the abundances of the tubulin family in cortical areas showed ascending trend from F50 to F90, which suggests this period may be critical for the formation of cerebral cortex. In contrast, a decreasing trend from F90 to P3 was observed, which indicates a decreasing proportion of neurons and an increasing proportion of non-neuronal cells during this period. As a member of ALB/AFP/VDB family, the AFP (alpha-fetoprotein) is abundant in brain during embryonic development and inhibits neurite growth and neuron migration by restraining the neurotrophic effect of oleic acid⁴⁶. We found that the abundance of AFP was continuously decreased from F50 to P3, suggesting that AFP may be an inverse indicator of neurodevelopment. As a member of protein kinases family, TAOK2 is essential for the formation and stability of dendritic spines as well as axon elongation in cortical neurons⁴⁷. Loss of TAOK2 activity causes autism-related neurodevelopmental and cognitive abnormality⁴⁸. In this study, TAOK2, with the correlation of 1 between monkey and human, showed a continuous increasing trend

from F50 to P3, which indicates that TAOK2 plays important role for the maintenance of normal neurodevelopment.” (Line 342-364)

We further performed cross-species comparison of the top 10 abundant protein families. We calculated the correlations for each protein in families among the five brain regions under the three pairwise comparisons (Mouse vs Human, Monkey vs Mouse, and Monkey vs Human). The results showed that each protein family presented proteins with both positive and negative correlations in either of the three pairwise comparisons. The monkey vs human comparison showed more proteins falling into the 0.75-1 correlation interval than those of the mouse vs human and monkey vs mouse. Although the overall similarity was high, we found the protein abundance of several specific members were distinguished between humans and monkeys. This result suggested the protein family were regulated in a species-specific manner. For example, in the tubulin family, TUBG1 and TUBA4A were highly correlated between monkey and human, while TUBB4A and TUBB6 were negatively correlated. In the protein kinase superfamily, several proteins such as TAOK2, PTK2, MAST1, MAP2K4, MAP2K1 have a correlation of 1 between monkey and human, while several proteins were negatively correlated. (Line 287-301, Fig. 6C and Fig. S6)

Furthermore, the study should also address which pathways are enriched in the categories where RNA and protein expression are conserved (type 4 and 5) when comparing the transcriptome and proteome results.

Response: Thank for your comments. As your suggestion, we have added the enrichment result for the genes from the type 4-6 categories in the revised manuscript. “Type 4-6 genes accounted for less than 5% of the all genes. Type4 genes showed consistent direction and fold change >2 at RNA level or at protein level. Type5 genes showed opposite direction between RNA level and in protein level. Type6 genes showed consistent direction and fold change >2 at both RNA level and protein level. Type4 genes were mainly enriched in the angiotensin related processes, lipid phosphorylation, coagulation, hemostasis, etc. Type5 genes were mainly enriched in the

cytoskeleton-related processes. Type6 genes were mainly enriched in protein localization to kinetochore, microvillus assembly, attachment of spindle microtubules to kinetochore, long-term synaptic depression, etc. In other brain regions, these six types of genes showed region-specific characteristic and are therefore enriched for different biological functions and processes (Fig S10-S11 and Table S10)” (Line 320-331, Fig S10-S11 and Table S10).

Additionally, the potential mechanisms underlying the discrepancy between RNA and protein expression and their biological relevance to brain development and evolution should be discussed.

Response: Thank for your comments. We discuss the potential mechanisms of the discrepancy between RNA and protein expression as:

“Our result revealed a certain proportion (>15%) of genes with mRNA/protein discrepancies. There are multiple reasons that could explain the discrepancy between RNA and protein expression⁵². The mRNA may be modified by methylation (e.g. m6a) and their stability and translational efficiency are dependent on these modifications. Protein stability also contributes the discrepancy between RNA and protein expression. Some protein has a long half-life while others are immediately destroyed for proper function. Gene transcription exists alternative splicing. Some splice variants are transcribed but not translated into protein. In addition, the miRNA and lncRNAs are regulatory factors that can control translation efficiencies.” (Line 423-431).

Reviewer #2 (Remarks to the Author):

The manuscript was modified substantially and i'm happy with the current version. I would still recommend removing Fig. 6A and the corresponding analysis. In my opinion it is an example of bad evolutionary analysis and Kunming is a place with good reputation for evolutionary work.

Response: Thanks for your appreciation of our work and the kindness reminding. As your suggest, we removed the results of Figure 6A in the revised manuscript.

Reviewer #3 (Remarks to the Author):

Big improvements have been added to the manuscript, new analysis and several new figures. All comments were addressed, and I think the updates improved the manuscript significantly. I would suggest going over the text once more, especially the new additions, to polish some formulations.

Response: Thanks for your appreciation of our work. We have carefully read the manuscript and checked it several times. And several mistakes were corrected.

I only have a couple minor comments:

Line 106, the phrase “protein expression” is still used to describe the protein detected by MS. The expression of proteins is usually described by mRNA i.e. transcriptomic. The comment from the authors says that they have substituted the use of protein expression, but I still see expression being used on several occasions. Using expression when talking about DEP makes sense, but make sure it is used consistently and with correct intention. Line 246 talks about quantified mean expression, but I assume we are still talking about protein detected? although the change in expression is what leads to change in protein abundance. in summary, stay consistent with the choice and make sure it is conscious.

Response: Thank for your suggestion. As the reviewer points out, we have rephrased the term “expression” with appropriate description. To be consistent and conscious, we use the term “abundance” to describe protein detection. For example, the sentences in Line 246 was revised as follows: “We next sought to determine abundance changes in the reported proteins implicated in the five diseases ¹². We quantified the mean abundance of these proteins between PFC and V1 at the four stages (Fig. 5D).” (**Line 253-254**).

line 118, “By analysis of proteins with subcellular localizations,” My assumption is that all proteins have a subcellular location, or extracellular. When I continue reading, I understand what you are referring to, but this sentence as an introduction provides no details what this is about. Additionally, why would you not find a high degree of similarity? Were you expecting the sub cellular location to be significantly different

between the regions? some type of buildup or hypothesis could improve this part of the story.

Response: Thank for your comments. We changed the description in line 118 as “The functions of cortex, subcortical region and CB are executed by different proteins distributed at various subcellular localizations. Therefore, we analyzed dynamic change of proteins with various subcellular localizations from F50 to P3 among brain regions. We found a high degree of similarity in the proportion changes of protein abundance within subcellular localizations among different regions of the cortex (Fig. 1E and Table S4).” (Line 118-123).

Line 124, “the known protein families”. Is that in comparison to the unknown protein families? This sentence could be reformulated.

Response: Thank for your comments. Here we mean the protein families that can be searched in the dataset. We changed the sentences as “In addition, we analyzed the patterns of spatiotemporal change of the protein families.” (Line 127).

Line 164-165 mentioning that there is an increased distance over development- that is a nice improvement of the CB story, great. Line 182, there are numerous publications related to the differentiation in development of granule cells, there must be some relevant knowledge related to the protein levels of GABRA6 to compare this result with instead of only speculating.

Response: Thank for your comments. As your suggestion, we compared our result with previous study. The revised contents as following. “The abundance of GRID2IP, PCP2 increased since F50, which indicates the proliferation or maturation of Purkinje cells at this stage. The abundance of ZIC1 increased since F50 and GABRA6 increased lately from F120. In human, granule cells were differentiated from progenitor cells at 10 weeks post conception, and the GABRA6 as a subunit of GABA-A receptor, is highly expressed in the postnatal CB, which mediates neuronal inhibition³⁰. The results suggest that the development of granule cells and the high expression of GABRA6 in the perinatal CB are similar between humans and monkeys. In addition, we analyzed

the changes of proteins (APC, CTNNB1, DDX3X, PTEN, BCOR) associated with medulloblastomas in CB at the four developmental stages.”(Line 183-192).

REVIEWERS' COMMENTS

Reviewer #1 (Remarks to the Author):

The manuscript has been significantly improved compared to the initial version. I am happy with the current version now.

Reviewer #3 (Remarks to the Author):

Nice improvements and the comments from the previous version have been taken care of. Im happy with the current version.

REVIEWERS' COMMENTS

Reviewer #1 (Remarks to the Author):

The manuscript has been significantly improved compared to the initial version. I am happy with the current version now.

Thanks for the comment.

Reviewer #3 (Remarks to the Author):

Nice improvements and the comments from the previous version have been taken care of. Im happy with the current version.

Thanks for the comment.